# Long-term measles antibody profiles following different vaccine schedules in China, a longitudinal study

Qianli Wang[1,9], Wei Wang[2,9], Amy K. Winter[3,9], Zhifei Zhan[4,9], Marco Ajelli [5], Filippo Trentini[6], Lili Wang[2], Fangcai Li[4], Juan Yang [2], Xingyu Xiang[4], Qiaohong Liao[2], Jiaxin Zhou[2], Jinxin Guo[2], Xuemei Yan[2], Nuolan Liu[2], C. Jessica E. Metcalf [7,8], Bryan T. Grenfell[7,8] & Hongjie Yu [1,2] ✉

Characterizing the long-term kinetics of maternally derived and vaccine-induced measles immunity is critical for informing measles immunization strategies moving forward. Based on two prospective cohorts of children in China, we estimate that maternally derived immunity against measles persists for 2.4 months. Following two-dose series of measles-containing vaccine (MCV) at 8 and 18 months of age, the immune protection against measles is not lifelong, and antibody concentrations are extrapolated to fall below the protective threshold of 200 mIU/ml at 14.3 years. A catch-up MCV dose in addition to the routine doses between 8 months and 5 years reduce the cumulative incidence of seroreversion by 79.3–88.7% by the age of 6 years. Our findings also support a good immune response after the first MCV vaccination at 8 months. These findings, coupled with the effectiveness of a catch-up dose in addition to the routine doses, could be instrumental to relevant stakeholders when planning routine immunization schedules and supplemental immunization activities.

Measles is a highly contagious virus that infects millions of children every year and caused more than 140,000 deaths globally in 2018[1,2]. It is estimated that the transmissibility of measles is the highest of any respiratory virus (basic reproduction number between 14 and 18)[2]. Although highly effective measles-containing vaccines (MCVs) have been recommended by the Expanded Programme on Immunization of the World Health Organization since 1974[3], measles immunity gaps remain. Gaps in vaccination, primary and secondary vaccine failure, vaccine hesitancy, and suspended or delayed immunization activities during the coronavirus disease 2019

(COVID-19) pandemic[4,5] have led to an increased risk of measles outbreaks worldwide[6].

The literature suggests that the offspring of women with vaccine-derived immunity lose their passive immunity faster than the offspring of women with immunity from natural infection[7]. Therefore, as an increasing number of girls are vaccinated globally, an increasing number of children will be born worldwide with shorter durations of maternally derived immunity[8]. The combination of more susceptible infants due to earlier loss of maternally derived immunity and an increasing risk of measles infections among children under five in

[1]Shanghai Institute of Infectious Disease and Biosecurity, Fudan University, Shanghai, China. [2]School of Public Health, Fudan University, Key Laboratory of Public Health Safety, Ministry of Education, Shanghai, China. [3]Department of Epidemiology and Biostatistics, University of Georgia, Athens, GA, USA. [4]Hunan Provincial Center for Disease Control and Prevention, Changsha, China. [5]Laboratory for Computational Epidemiology and Public Health, Department of Epidemiology and Biostatistics, Indiana University School of Public Health, Bloomington, IN, USA. [6]Dondena Centre for Research on Social Dynamics and Public Policy, Bocconi University, Milan, Italy. [7]Department of Ecology and Evolutionary Biology, Princeton University, Princeton, NJ, USA. [8]Princeton School of Public and International Affairs, Princeton University, Princeton, NJ, USA. [9]These authors contributed equally: Qianli Wang, Wei Wang, Amy K Winter, Zhifei Zhan. ✉e-mail: yhj@fudan.edu.cn

China and worldwide could lead to large outbreaks among the most vulnerable populations. This may be particularly true in countries such as China where an increased risk of measles outbreaks can be expected because of the combined effects of a continuous decline in population immunity (declining from 94.7% in 2009 to 75.4% in 2015)[9] and measles importation risk[10]. As a result, the recommended age for MCV dose 1 (MCV1) at 9 months in high-risk countries and 11–15 months in low-risk countries, which has been established in accordance with the decay of maternally derived antibodies, may need to be reconsidered. Although studies that focus on the kinetics of maternal antibodies do exist, they tend to be geographically limited and may not be generalizable to today's Chinese populations given region- and time-specific changes in virus circulation, demography, and vaccine coverage.

To optimize the target age of MCV1, one must also consider age-specific vaccine effectiveness. Knowledge gaps remain concerning whether administering MCV1 to infants at an age earlier than 8 months can induce long-term protective immune responses against measles virus infection without interfering with the immune response to subsequent MCV doses[11,12]. Although two recent meta-analyses suggest good immunogenicity of MCV1 vaccination in infants younger than 9 months[13,14], no study has considered the long-term kinetics of vaccine-induced immune response after MCV1 vaccination at an age younger than 9 months. Therefore, there is an urgent need to obtain insights into the kinetics of maternally derived immunity and vaccine-derived immunity in children with MCV1 at an age younger than 9 months to inform the target age for routine MCV doses and to establish or refine catch-up schedules.

China's childhood immunization schedule includes two doses of routine vaccination at ages 8 and 18 months, which was developed in accordance with infant immune system development, the age distribution of measles cases, and the levels of maternal and vaccine-induced immunity against measles at a population level[15]. Supplementary immunization activities that target children between the ages of 8 months and 16 years were conducted intermittently between 2003 and 2018. Representative longitudinal serological surveys conducted among Chinese children allow us to simultaneously evaluate the kinetics of measles maternal antibodies and antibodies following vaccination. We describe the long-term average immunoglobulin G (IgG) antibody concentrations prior to and following different vaccination schedules against measles, and evaluate the reduction in the cumulative incidence of seroreversion due to a catch-up dose in addition to the routine doses. Our results can be used to inform the current routine vaccination schedule against measles for Chinese children and the utility of a catch-up dose in addition to the routine doses in maintaining immunity in the long run.

## Results

We enroled 2629 children in this study (Fig. 1); 555, 352, 354, 318, 335, 316, and 399 children were enroled at birth and at 1, 2, 3, 4, 5, and 6–9 years of age, respectively. Of these, 1268 (48.2%) children were female (Table 1). The follow-up time ranged from 1 to 42 months (median 35.2 months, interquartile range (IQR) 29.3–36.2). The epidemic curve of measles for our study locations is presented in Supplementary Fig. 1. Fewer than 150 cases were reported annually, and no study participant reported a history of measles infection.

Limited by the availability of vaccination cards and participant compliance, immunization records were only recorded for 1741 (66.2%) study participants. Among these, 310 children had no record of the timing of receiving each dose of MCV. Among 1431 children with complete immunization records, 834 (58.3%) children received MCV1 at 8 months (Supplementary Fig. 2), and 300 (21.0%) children received two routine doses at ages 8 and 18 months. Supplementary Fig. 3A shows the exact time of administration of two routine MCV doses and blood sampling for these participants. Among 565 children who were involved in catch-up immunization activities, 387 (68.5%) children

received two routine MCV doses and an additional catch-up dose between 8 and 18 months; and 34 (6.0%) children received two routine MCV doses and an additional catch-up dose between 2 and 5 years. Slight differences in age structures between children with ($n = 1741$) and without ($n = 888$) vaccination records were observed (Supplementary Table S1), while the observed IgG antibody concentrations between children with and without vaccination records were consistent across age groups (Supplementary Fig. 3B).

It is important to identify factors associated with children's measles-specific IgG antibodies before fitting the kinetic curves of antibody concentrations by age. We thus performed a multivariate regression analysis with a generalized linear mixed model (GLMM), which found only two statistically significant factors associated with measles-specific IgG antibodies in children: age and the number of MCV doses (MCV1 vs. non-vaccination: $p$-value < 0.001; MCV2 vs. non-vaccination: $p$-value < 0.001) (Supplementary Table S2). In particular, the IgG antibody concentration decreased with age ($p$-value < 0.001). We found that children's baseline characteristics, age and vaccination status together accounted for 32.3% of the total variance (Supplementary Table S3). Furthermore, although individual heterogeneities in measles-specific immune response were relatively low (7.8% of the total variance), random effects that allowed to account for individual heterogeneities in immune response to MCV were introduced in the kinetic model of antibody decay (Supplementary Table S4).

Individual IgG antibody concentration trajectories before and after undergoing different vaccination schedules are presented in Supplementary Fig. 4. Maternally derived antibody concentrations in infants diminished sharply in the first six months of life, as apparent in IgG geometric mean concentrations (GMCs) by age group (Fig. 2). We observed that the log-transformed GMCs decreased from 6.8 (95% confidence interval (CI) 6.7–7.0) at 0–1 months, to 3.4 (95% CI 3.4–3.5) at 6–7 months (Fig. 2A). According to the generalized additive mixed models, the mean time for IgG antibody concentrations to fall below the protective threshold of 200 mIU/ml was 2.4 months (Fig. 2B). At the age of 8.0 months, log-transformed IgG antibody concentrations reached the lowest value of 2.8. Our results show that the choice of the protective threshold did not substantially change the time for IgG antibody concentrations to fall below the protective threshold (3.6 and 1.9 months for 120 mIU/ml and 300 mIU/ml thresholds, respectively) and to reach the lowest level of IgG antibody concentration (8.0 months for both 120 mIU/ml and 300 mIU/ml thresholds) (Supplementary Fig. 5).

Since failure to seroconvert after vaccination is a separate dynamic from vaccine-induced seroconversion followed by waning of antibody concentrations, in the main analysis, we reported MCV-induced antibody response and persistence by excluding individuals who failed to seroconvert after MCV vaccination ($n = 5$) (individual trajectories are shown in Supplementary Fig. S4). The Appendix also shows a sensitivity analysis where these observations are not excluded (Supplementary Figs. S6 and S7). We found that, following MCV1 vaccination at 8 months of age, log-transformed IgG antibody concentrations gradually increased from 2.8 at 8.0 months to 7.3 at 9.0 months (Fig. 2). Using paired serum samples from infants at 6 and 24 months of age ($n = 69$), the proportion of children with seroconversion or a fourfold rise in IgG antibody following MCV1 at 8 months was estimated to be 98.6% (95% CI 92.2–99.9%). By extrapolating generalized additive mixed model (GAMM) fitted parameter values beyond the observed 24 months of age, it was estimated that IgG antibody concentrations would drop below the protective threshold of 200 mIU/ml at 53 months post MCV1 administration (i.e., 5.1 years of age) (Supplementary Fig. 8A).

For participants receiving the second routine dose of MCV (i.e., MCV2) at 18 months of age, we found that IgG antibody concentrations peaked one month following vaccination (3378 mIU/ml at 19.0 months), and IgG antibody concentrations remained above

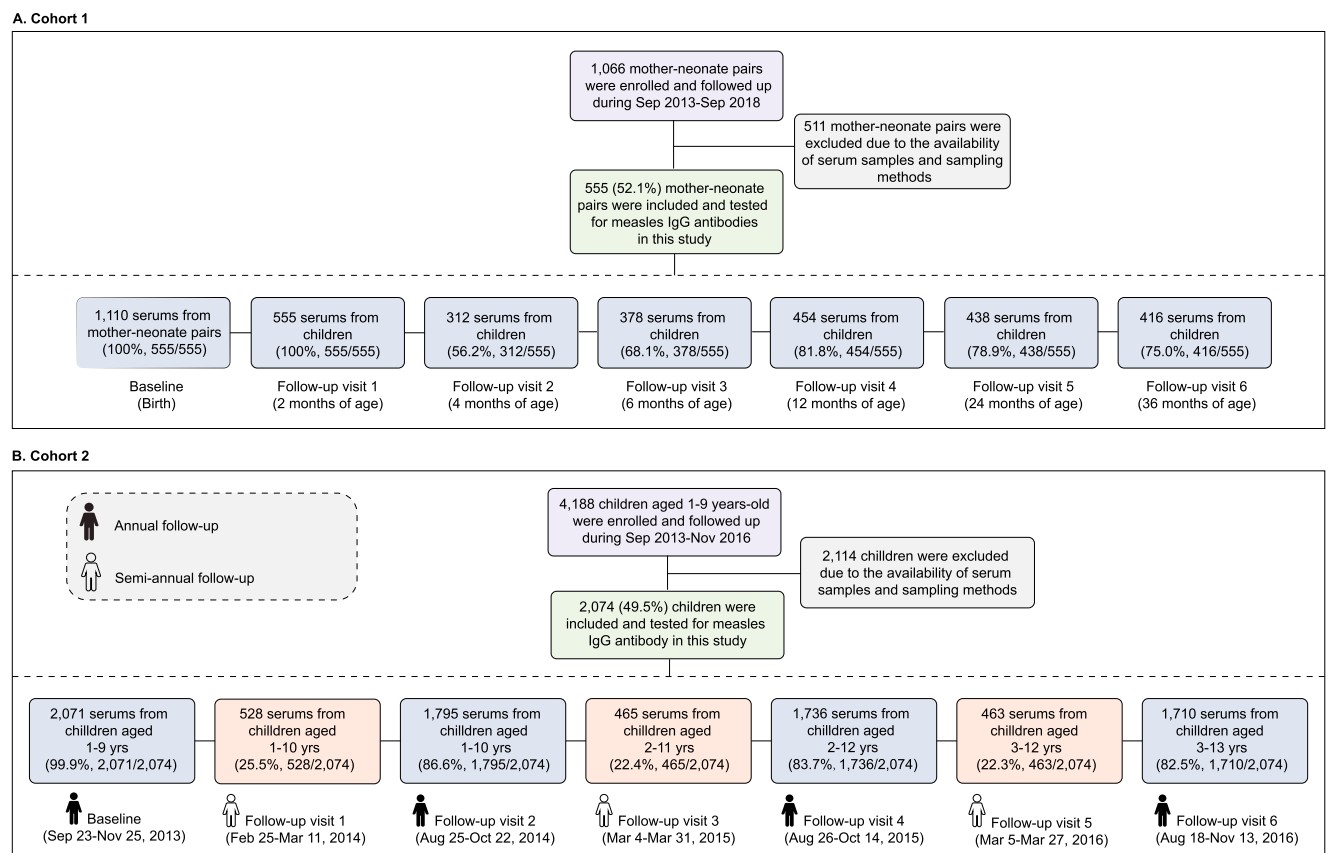

**Fig. 1 | Overview of study participants. A** Recruitment and follow-up of participants in the cohort of mother–neonate pairs and **B** in the cohort of children aged 1–9 years.

the protective threshold of 200 mIU/ml for the remaining follow-up visits (approximately 9.4 years after MCV2) (Fig. 3). Moreover, the simulated evolution of IgG antibody concentrations showed that, 12.8 years after MCV2 vaccination (i.e., 14.3 years of age), IgG antibodies would drop below the protective threshold (Supplementary Fig. 8B).

Age-specific IgG GMCs among children with and without a catch-up dose in addition to the routine doses (i.e., the routine and catch-up vaccination groups) are shown in Supplementary Fig. 9. Although IgG GMCs declined with age in both vaccination groups, IgG GMCs were generally higher in the catch-up vaccination group than in the routine vaccination group within the same age group, regardless of age at catch-up vaccination (Supplementary Fig. 9). We also found that, since the third year of the child's life, their cumulative incidence of seroreversion increased continuously with age (Fig. 4). And such increasing age-specific incidence of seroreversion was not affected by the impact of the enzyme-linked immunosorbent assay (ELISA) on detecting near-threshold values when using a consistency rate of 95.8% between concentrations by neutralization and ELISA (Supplementary Fig. 10). Moreover, the cumulative incidence of seroreversion in the routine vaccination group was significantly higher than those in the catch-up vaccination group, regardless of which protective threshold was used (p-value < 0.001, Fig. 4, Supplementary Fig. S11). In particular, by the age of 6 years, a catch-up dose between 8 and 18 months of age showed a 79.3% (95% CI 70.5–100%) reduction in the cumulative incidence of seroreversion, and a catch-up dose between 2 and 5 years of age was associated with an 88.7% (95%CI 83.5–100%) reduction by assuming a protective threshold of 200 mIU/ml (Fig. 4).

## Discussion

We have provided an assessment of the rapidly changing kinetics of maternally derived and vaccine-induced antibodies in Chinese children from birth to 11 years of age. We estimated that by 6 months of age, IgG antibody concentrations fall below thresholds believed to be protective against wild virus infection (200 mIU/ml[16–18]) and to interfere with vaccine response (50 mIU/ml[19,20]). We also found that it is unlikely that a single dose of MCV1 at 8 months of age can provide long-term protective immunity, but it might be enough to protect children from viral infection before 5 years of age. In a population of children who received 2 doses of measles vaccine at ages consistent with the recommendation of China's national immunization program (NIP) (i.e., 8 and 18 months), the IgG antibody concentration fell below protective levels at 14.3 years of age. In addition, we observed that a catch-up dose in addition to the routine doses between ages 8 months and 5 years have substantially reduced individuals' cumulative incidence of seroreversion before 6 years of age, regardless of age at catch-up vaccination. These two findings indicate that early vaccination at 8 months of age would not interfere with the immune response to subsequent MCV doses.

Our study found that maternal antibody concentrations decreased below the protective threshold within 1–2 months of life (half-life of 2.3 months), which is consistent with the findings by Leuridan et al.[7]. We therefore infer that interference from maternal antibodies would no longer be a barrier to vaccine immunogenicity by six months of age in China, when the lowest IgG antibody level is reached. The very low fraction of protected individuals at 6–7 months (3.5%, 360/373) is compatible with the levels found in unvaccinated infants at 9 months in Burkina Faso and Guinea-Bissau (2.8–4.8%), which have higher rates of measles transmission[21]. This provides evidence that the waning rate of maternally derived antibodies may be faster if vaccinated mothers' immune response against measles has not been boosted by natural exposure (as often occurs in an elimination setting). The above findings, coupled with recent evidence regarding

**Table 1 | General characteristics of the children**

| Characteristics | Total (n = 2629) | Neonates (n = 555) | Children aged 1–9 years (n = 2074) |
|---|---|---|---|
| Age at baseline, months or years | | | |
| Median (interquartile range, IQR, years) | 3.1 (1.4–5.1) | — | 4.0 (2.4–5.5) |
| 0–7 months | 555 (21.1) | 555 (100) | 0 (0) |
| 8–17 months | 121 (4.6) | — | 121 (5.8) |
| 18–24 months | 236 (9.0) | — | 236 (11.4) |
| 25 months–6 years | 1336 (50.8) | — | 1336 (64.4) |
| 7–10 years | 381 (14.5) | — | 381 (18.4) |
| Sex | | | |
| Female | 1268 (48.2) | 254 (45.8) | 1014 (48.9) |
| Male | 1361 (51.8) | 301 (54.2) | 1060 (51.1) |
| Gestational age, weeks | | | |
| Median, IQR | — | 40 (39.1–40.7) | — |
| Preterm birth (<37) | 142 (5.4) | 36 (6.5) | 106 (5.1) |
| Full-term birth | 2451 (93.2) | 492 (88.6) | 1959 (94.5) |
| Post-term birth (≥42) | 36 (1.4) | 27 (4.9) | 9 (0.4) |
| Mode of delivery (n, %) | | | |
| Vaginal delivery | 1673 (63.6) | 349 (62.9) | 1324 (63.8) |
| Caesarean section | 956 (36.4) | 206 (37.1) | 750 (36.2) |
| Birth weight, grams | | | |
| Median, IQR | 3250 (3000–3500) | 3300 (3000–3600) | 3250 (3000–3500) |
| <2500 | 85 (3.2) | 9 (1.6) | 76 (3.7) |
| 2500 to <4000 | 2301 (87.5) | 509 (91.7) | 1792 (86.4) |
| ≥4000 | 243 (9.2) | 37 (6.7) | 206 (9.9) |
| Breastfeeding before 6 months | | | |
| Yes | 2309 (87.8) | 485 (87.4) | 1824 (87.9) |
| No | 319 (12.1) | 69 (12.4) | 250 (12.1) |
| Missing | 1 (0.1) | 1 (0.2) | 0 (0) |
| Socioeconomic status[a] | | | |
| Low | 580 (22.1) | 97 (17.5) | 483 (23.3) |
| Middle | 1250 (47.5) | 95 (17.1) | 1155 (55.7) |
| High | 589 (22.4) | 153 (27.6) | 436 (21.0) |
| Missing | 210 (8.0) | 210 (37.8) | 0 (0) |
| Vaccination against measles during follow-up visits (n, %) | | | |
| No. (%) of children with one dose | 36 (1.4) | 25 (4.5) | 11 (0.5) |
| No. (%) of children with two doses | 1070 (40.7) | 391 (70.5) | 679 (32.7) |
| No. (%) of children with three or more doses | 635 (24.2) | 2 (0.4) | 633 (30.5) |
| Unknown | 888 (33.8) | 137 (24.7) | 751 (36.2) |

[a]The socioeconomic status index is calculated using annual household income given data availability.

stronger beneficial effects of early MCV1 vaccination on children's protective immunity and on their survival[22], as well as the increasing risk of infection among children below the age of scheduled vaccination[23], highlight the potential for lowering the MCV1 target to under 8 months of age in China, particularly in a scenario in which measles outbreaks continue to occur. Nonetheless, there are important considerations before lowering the MCV1 target age in China and other countries where children under 5 years of age are at high risk of measles infection. First, despite a possible high MCV1 coverage among infants younger than 9 months after lowering the MCV1 target age, local health authorities should be aware of the quicker waning of antibodies in children who received MCV1 before 9 months of age compared to those who received the dose at 9 months of age or older[24,25]. Moreover, the role of cell-mediated immunity should also be

considered. Jaye and colleagues[26,27] found that the absence of detectable/protective antibodies may not directly translate into susceptibility to infection and disease severity, possibly due to T-cell immunity.

Following MCV1 administration at 8 months, extrapolated IgG antibody concentration levels remained above a protective threshold of 200 mIU/ml until 53.0 months post MCV1, suggesting a target age for MCV2 at approximately 4 years of age. This finding was consistent with Brinkman et al.'s recent study, in which they demonstrated that neutralizing antibody levels dropped below the cut-off for clinical protection at 60 months post MCV1 administration at age 6–8 months[24]. Although a 53.0-month antibody persistence after MCV1 at 8 months was much shorter than that reported in a systematic review of MCV persistence (approximately 35 years post-MCV1)[28], the difference could be explained by a different measles infection risk at the time of our study with respect to other studies included in the meta-analysis. In Appendix we show that the duration of protection following MCV1 vaccination is sensitive to three observations at 26–27 months, which we excluded from the main analysis due to a lack of information about their status after MCV1 vaccination. Further modelling work is needed to reconstruct individual immune response profiles against measles following different combinations of MCV1 and MCV2 schedules to determine context-specific optimal schedules for MCV vaccination.

The present study suggested that the seroconversion rate of 98.6% among cohort participants with MCV1 at 8 months is close to that among children receiving MCV1 at 12 months or older (83–98%)[29]. Moreover, we found that the cumulative incidence of seroreversion was significantly lower among children with two routine MCV doses plus one catch-up dose than among those with two routine MCV doses only. These findings suggest that the blunting of immune responses after MCV1 at 8 months, which was found by Nic and colleagues[14], may be more limited than before (a seroconversion rate of 60–70%[29]), particularly after increasing the number of MCV doses. However, the seroconversion rate following MCV1 at 8 months and the net effect of MCV1 at 8 months on the response to the subsequent MCV dose need to be quantified using large population studies. In addition, despite the effectiveness of a third MCV dose, catch-up immunization should prioritize children who missed routine vaccination rather than targeting fully vaccinated children when the primary objective is to increase MCV coverage for measles outbreak prevention.

A few limitations of this study should be highlighted. First, ELISA does not distinguish between functional and nonfunctional antibodies, and it may tend to overestimate equivocal and negative results. However, our analyses of the ability of ELISA to detect near-threshold values suggest that ELISA would not affect our key findings, consistent with previous studies[30,31] showing that ELISA is adequate to test immunity and identify seronegative individuals. Second, although there is some debate regarding the level of the protective threshold ELISA value for measles, this does not impact our findings regarding the early waning of maternally derived antibodies, as no substantial difference was observed in the estimated mean time to decrease below the protective threshold when we adopted alternative protective thresholds[7,32]. Third, the fraction of individuals with vaccination records was relatively low, but this would not limit the generalizability of our findings given that antibody kinetics from birth to the end of the follow-up were similar between participants with and without vaccination cards. Fourth, the impact of measles outbreaks on antibody levels should be taken into consideration, especially as it significantly affects both maternal and vaccine-induced antibodies. Unfortunately, we did not have representative paired mother-neonate samples before and after a measles outbreak; thus, we could not adjust for this factor, possibly leading to the overestimation of vaccine-induced antibody concentrations. Fifth, the GAMM model used in this study tended to smooth observations in each age group, and failed to account for

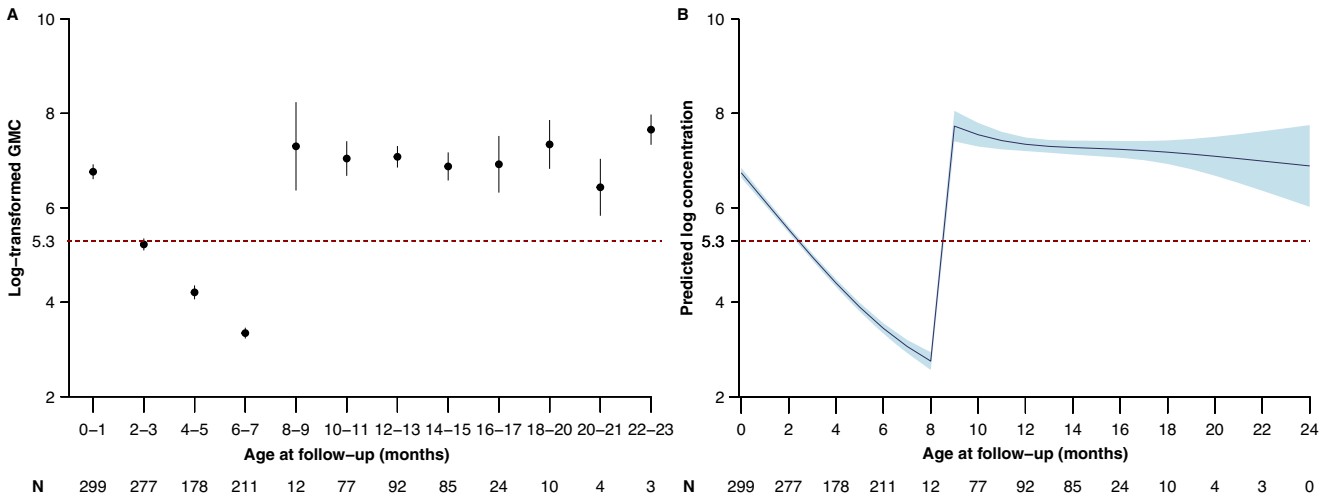

**Fig. 2 | Fitted profiles for measles antibody concentrations among children with the first dose of MCV at 8 months. A** Observed and **B** predicted log-transformed geometric mean concentrations. The points in **A** refer to the log-transformed geometric mean concentrations. The thick curve in **B** is the predicted mean values of log-transformed concentrations from a generalized additive mixed model. Error bars and shaded areas show 95% confidence intervals. Horizontal red dashed lines refer to a protective threshold of 200 mIU/ml. In the bottom panel, the letter N refer to the sample size used to derive the error bars and shaded areas.

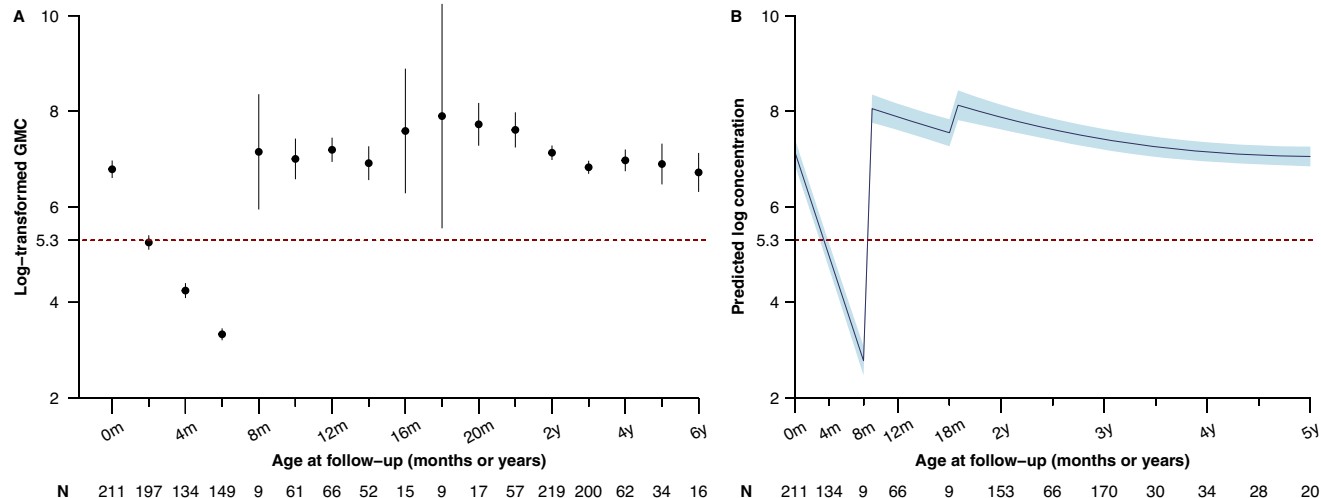

**Fig. 3 | Fitted profiles for measles antibody concentrations among children with a two-dose routine vaccination schedule consistent with the recommendation of China's national immunization program. A** Observed and **B** predicted log-transformed geometric mean concentrations. The points in **A** refer to the log-transformed geometric mean concentrations. The thick curve in **B** is the predicted mean values of log-transformed concentrations from a generalized additive mixed model. Error bars and shaded areas show 95% confidence intervals. Horizontal red dashed lines refer to a protective threshold of 200 mIU/ml. In the bottom panel, the letter N refer to the sample size used to derive the error bars and shaded areas.

individual antibody concentration trajectories, particularly for those who had a low response to MCV. This might directly lead to an underestimated IgG antibody concentration and persistence following different vaccination schedules. Moreover, as none of the children with seropositive serum samples after MCV1 vaccination at 8 months of age had shown evidence of seroreversion by the time of their MCV2 vaccination, we were unable to characterize the long-term susceptibility profiles following a single-dose of MCV at ages 8 months using the Kaplan-Meier method. More individuals with consecutive serum samples before and after each MCV vaccination should be involved to reconstruct individual immune response profiles against measles prior to and following different vaccination schedules. Additionally, restricting the study to participants who had received 2 doses of measles vaccine at ages consistent with the NIP recommendation did not allow us to shed light on the differences in vaccine-induced immune responses in sizable cohorts of children with MCV1 and MCV2 vaccination at different ages. Further studies are needed to capture the

antibody dynamics following different combinations of MCV1 and MCV2 vaccination schedules.

In conclusion, our findings suggest that targeting individuals under 8 months of age for MCV1 is necessary to reduce the number of susceptible Chinese children below the age of the currently scheduled vaccination. While administering MCV1 at 8 months of age cannot provide long-term protective immunity for children, it also does not interfere with the responses to subsequent MCV doses. Moreover, the two-dose routine vaccination schedule recommended in China can effectively protect children during their childhood and adolescence. Further serological studies are needed to assess whether measles antibody concentrations remain above the protective threshold during adulthood, especially if the target age for MCV1 is lowered in a scenario in which measles outbreaks continue to occur. The estimated reduction in the cumulative incidence of seroreversion due to a catch-up dose in addition to the routine doses suggests the effectiveness of nonselective supplementary immunization activities against measles

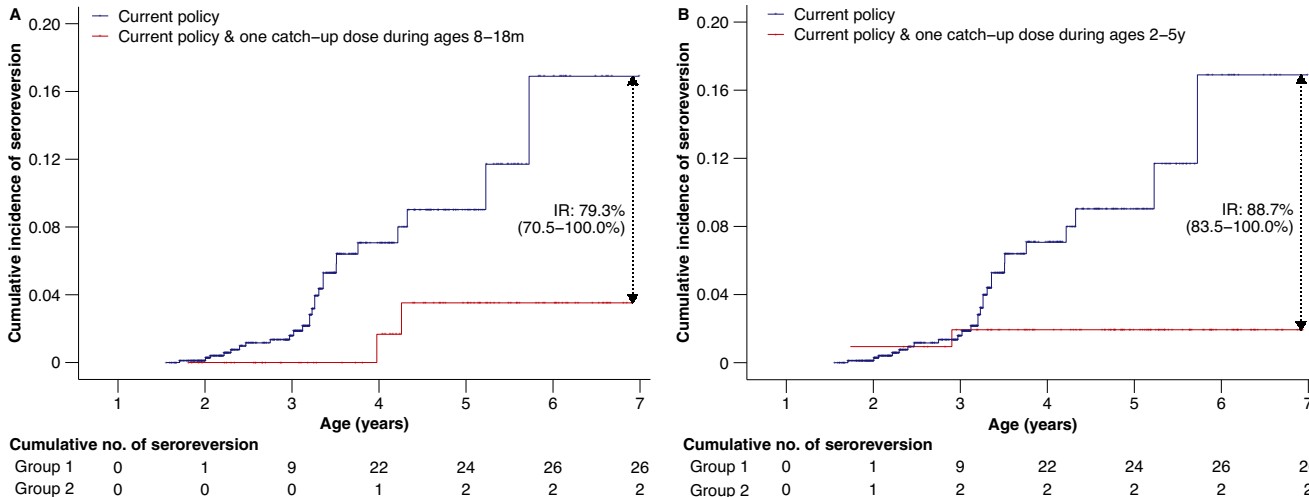

**Fig. 4 | Cumulative incidence of seroreversion among children receiving the first dose of MCV at 8 months of age and subsequent one or two doses between the ages of 8 months and 5 years, using a protective threshold of 200 mIU/ml.** **A** Cumulative incidence of seroreversion among children who received two routine MCV doses at 8 and 18 months of age (i.e., current policy) or received two routine MCV doses and a catch-up dose between 8 and 18 months of age. **B** Cumulative incidence of seroreversion among children who received two routine MCV doses at 8 and 18 months of age (i.e., current policy) or received two routine MCV doses and a catch-up dose between 2 and 5 years of age. In the bottom panel, Group 1 represents children who received two routine MCV doses at 8 and 18 months; Group 2 represents children who received two routine MCV doses and a catch-up dose between the ages of 8 months and 5 years. The abbreviation "IR" indicates the reduction in the cumulative incidence of seroreversion between the two groups.

implemented in China, which can be considered a prioritized intervention for reducing both the number of susceptible individuals and of infections in children younger than 5 years of age. Given the profoundly negative effects of the COVID-19 pandemic on childhood immunization coverage, the above findings can guide relevant stakeholders in planning and implementing nationwide or regional catch-up measles vaccination strategies after the pandemic.

## Methods

### Study design and participants
We used archived serum samples from two community-based longitudinal cohorts, including a cohort of mother-neonate pairs ($n = 1066$) and a cohort of children aged 1-9 years ($n = 4188$), that aimed to investigate the sero-epidemiological characteristics of paediatric enterovirus A71 infections in Hunan Province, China between September 2013 and September 2018. Detailed information on the cohort profile and on the collection of baseline characteristics and vaccination cards for cohort children is provided in Appendix. Only a subsample of the original cohorts was included in this study. In particular, we tested 2629 participants' serum samples for measles-specific IgG antibody; the included sera came from 555 mother-neonate pairs and 2074 children aged between 1 and 9 years (Fig. 1). We categorized these cohort children into four vaccination groups: (a) children who received MCV1 vaccination only at age 8 months; (b) children who received two routine MCV doses (i.e., MCV1 and MCV2) at ages 8 and 18 months; (c) children who received two routine MCV doses and an additional catch-up dose between 8 and 18 months of age; and (d) children who received two routine MCV doses and an additional catch-up dose between 2 and 5 years of age.

### Laboratory procedures
Quantitative results of measles-specific IgG antibody were obtained by using commercial ELISA kits (SERION ELISA classic measles virus IgG, Institut Virion/Serion GmbH, Wurzburg, Germany, see details in Appendix). Given that a protective threshold ELISA value for measles has not been established, we first used a generally accepted protective threshold of 200 mIU/ml[16,17] in the main analysis and then explored two additional values (120[32] and 300 mIU/ml[7]) in sensitivity analyses, where

300 mIU/ml corresponded to the upper limit assumed by Leuridan and colleagues[7]. Samples with equivocal results (150–200 mIU/ml) were considered negative in the main analysis. We considered participants whose antibody concentrations were below the protective threshold to be susceptible individuals (i.e., seronegative individuals), while protected or immune individuals were defined as those with antibody concentrations greater than the protective threshold (i.e., seropositive individuals). Seroconversion was defined as a change from seronegativity to seropositivity at time points before and after MCV1 vaccination, whereas seroreversion indicated the loss of protective measles-specific antibody concentrations after vaccination.

Furthermore, to evaluate the consistency between antibody concentrations measured through ELISA and the "gold standard" plaque reduction neutralization test (PRNT), a subset of 120 serum samples (including positive, equivocal, and negative IgG ELISA results) were re-evaluated using the PRNT. We adjusted the results obtained with ELISA in detecting near-threshold values (defined here as values between a protective threshold ± 50 mIU/ml)[33] by use of a concordance rate of 95.8% between ELISA and the PRNT (see the details in Appendix).

### Statistical analyses
We conducted descriptive analyses to evaluate the baseline characteristics of the participants. To describe the age distribution of cohort participants and measles antibody kinetics prior to or following measles vaccination for the four vaccination groups of children (described above), we combined data from baseline and follow-up visits from the two study cohorts by the age of the child. We first grouped children into 5 age groups to characterize their age distribution, and then grouped them into 17 age groups (i.e., two months up to 24 months and then by 1 year up to 5 years) to describe measles antibody kinetics.

We characterized IgG antibody geometric mean concentrations (GMCs) across age groups, along with 95% confidence intervals (CIs). Moreover, we calculated the seroconversion rate among infants who received MCV1 at 8 months. IgG antibody concentrations were log-transformed, and GMCs were compared using the Wilcoxon signed-rank test. We considered a $p$-value <0.05 to be significant.

A generalized linear mixed-effects model (GLMM) was built to quantify the effects of potential factors associated with IgG antibodies (Appendix, p7). Incorporating children's age and vaccination status (including pre-vaccination, after a single dose vaccination at 8 months, or after two-dose vaccination at 8 and 18 months), we then used generalized additive mixed models (GAMMs) to fit the log of IgG antibody concentrations at different ages among children who seroconverted after either one dose of MCV1 or the routine two-dose MCV (Appendix, p8). In addition, we performed two sensitivity analyses where we considered the uncertainties in the estimated persistence of MCV-induced antibodies due to individual heterogeneities in the immune response to MCV. In the first sensitivity analysis, we removed random effects in GAMM model; in the second analysis, we included observations from children who failed to seroconvert after MCV vaccination. These sensitivity analyses are reported in Supplementary Figures S6 and S7 in Appendix. To determine the impact of the individual heterogeneities on MCV1-induced immune response, the immune response to MCV1 was fitted using repeatedly drawn bootstrap samples from all observations or from observations that excluded the seronegative child at 26–27 months who failed to seroconvert after MCV1 vaccination (Appendix, p18).

To differentiate whether a catch-up dose in addition to the routine doses was potentially necessary to maintain individual and population immunity in the long run, Kaplan–Meier survival methods were used to calculate the cumulative incidence of seroreversion in IgG antibodies by a given age. For the Kaplan–Meier analyses, we used a threshold of 200 mIU/ml to define seroreverted individuals in the main analyses, whereas 300 mIU/ml was selected as an alternative threshold in the sensitivity analysis. We then compared the estimated cumulative incidence of seroreversion by age 6 years among children who received two routine MCV doses only and those who, in addition to the routine doses, received a catch-up dose between the ages of 8 months and 5 years. The log-rank test was used to compare the cumulative incidences between different vaccination groups. All analyses were conducted with R statistical software, version 4.1.0[34]. The data were stored and maintained using Microsoft Office Excel 2019.

### Ethical approval statement
This study was approved by the Institutional Review Board of WHO Western Pacific Regional Office (2013.10.CHN.2.ESR), the Chinese Centre for Disease Control and Prevention (201224), and Fudan University (2019–05–0756), and written informed consent was obtained from all caregivers of participants.

### Role of the funding source
The funders had no role in the design or conduct of this study; collection, management, analysis, or interpretation of the data; preparation, review, or approval of the manuscript; or decision to submit the manuscript for publication.

### Reporting summary
Further information on research design is available in the Nature Portfolio Reporting Summary linked to this article.

## Data availability
The data generated in this study have been deposited in the repository under https://github.com/Sueleaf/antibody-dynamics-against-measles (https://doi.org/10.5281/zenodo.7676630).

## Code availability
The code to reproduce the results and plots of this study have been deposited in the repository under https://github.com/Sueleaf/antibody- dynamics-against-measles (https://doi.org/10.5281/zenodo. 7676630).

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

## Acknowledgements

H.Y. acknowledges financial support from the Key Program of the National Natural Science Foundation of China (No. 82130093). W.W. acknowledges financial support from the National Postdoctoral Program for Innovative Talent (No. BX2021072) and the China Postdoctoral Science Foundation (No. 2022M720753). The authors thank Prof. Judy Beeler and Susette Audet from the Division of Viral Products in U.S. Food and Drug Administration for technical guidance in performing PRNT assays. We also thank Dr. Maria Litvinova in Indiana University School of Public Health for technical supports in building statistical models to estimate the immune response prior and following different vaccination strategies.

## Author contributions

H.Y. conceived, designed and supervised the study. Q.W., Z.Z., F.L., J.Y., Q.L., X.X., J.Z., J.G., X.Y., and N.L. participated in data collection. Q.W., L.W., and J.G. did the laboratory tests. W.W., Q.W., X.Y., J.Z. and N.L. analyzed the data. Q.W., W.W., and A.W. prepared the figures and the first draft of the manuscript. H.Y., A.W., Z.Z., M.A., F.T., B.G., and J.M. commented on the data and its interpretation and revised the content critically. All authors contributed to the review and revision, approved the final manuscript as submitted and agree to be accountable for all aspects of the work.

## Competing interests

H.Y. has received research funding from Sanofi Pasteur, and Shanghai Roche Pharmaceutical Company. M.A. has received research funding from Seqirus. None of those research funding is related to measles. All other authors report no competing interests.
