## [Peer Review File · Nature Communications]

Long-term measles antibody profiles following different vaccine schedules in China, a longitudinal studyEditorial Note: Parts of this Peer Review File have been redacted as indicated to maintain the confidentiality of unpublished data.

REVIEWER COMMENTS

Reviewer #1 (Remarks to the Author):

Significance

Overall, the manuscript aims to address an important topic of major public health interest – the timing of routine measles vaccination of infants, in an era and setting where maternally-derived protection is waning faster due to a cohort of mothers whose immunity is largely vaccine-derived rather than from natural infection and who have not faced repeated challenge by the measles virus as they live in a near-elimination setting. This is a critical decision, trading off the risk of measles infection in infancy vs. the potential for suboptimal vaccine response and thus infection later in life, and there remains a good deal of uncertainty about the magnitude and durability of the antibody response to earlier MCV doses. The authors also have a cohort of children who have received 3 measles vaccine doses, which is not recommended in any schedule that I'm aware of, but I at least believe is an open question as to whether a third dose might potentially be necessary to maintaining population immunity over long timescales in elimination settings. The authors have identified a dataset that appears very promising in addressing the uncertainty around these questions.

Key Results

The manuscript describes a longitudinal dataset of serum samples, with a cohort of neonates and a cohort of older children 1-10 years of age, followed up over three years. The samples were tested for measles IgG antibody, and on the basis of this data the manuscript makes four central claims:

1. Waning of maternally-derived immunity with a half-life of about 2 months
2. Immunity from a single dose of MCV wanes by 26 months of age
3. Two MCV doses at 8 and 18 months provide high levels of population immunity, stable through at least 9 years of age
4. A catchup dose between 8 months and 5 years reduces a child's relative risk of becoming susceptible (clarity needed in the statement of this claim – becoming susceptible by what age?)

Validity

I find the basis for two of these key results very unconvincing in the current analysis. here are also a number of statements that appear to be internally inconsistent

Waning of maternally-derived immunity with a half-life of about 2 months: very clearly supported by the data in Figs. 2A/B

Waning of single-dose immunity by 26 months of age: Very unconvincing. Looking at Figures 2A and B, the data presented here seem to me to provide thin support for making such a strong claim about antibody waning. The finding appears to be driven by a single seronegative, out of three samples, in the final age bin at 26-27 months (in discussion, at lines 246-247, the authors say it is driven by three individuals who were categorized as seronegative at 26-27 months, but Figure 2B indicates that there are 3 total individuals at 26-27 months, only 1 of whom is seronegative).

To me, this seems like an artifact of a model given too much freedom to curve sharply to match a single data point at the edge of the fit range (Fig. 2D). I'm curious what the authors would find if they re-fit the % susceptible model under repeated bootstrapping of the data; I would guess that, given the lack of any susceptibles found in the 18-23 month age bins, the duration of protection bounces from 26 months to lifelong based on whether that 1 seronegative is in the bootstrap sample or not.

There are other causes for concern about this result. The model output presented in Fig 2D has unrealistically low variance on the proportion susceptible in the 15-22 month age bin given the error bars on the data shown in Fig. 2B; the model really overstates confidence here. And there are distinctly internally inconsistent statements being made in this part of the results – at line 157: "susceptibility to infection decreased ... to 1.1% at 24 months", but at 159 "seroconversion rate following MCV1 at 8 months is estimated to be 89.6%". That's a full order of magnitude difference in effectiveness. Similarly, 1.1% susceptibility at 24 months is quite hard to square with a duration of protection of only 26 months (with a 95% interval down to 23.6 months); this would require remarkably rapid and synchronous waning across a population showing a lot of variance in Ab concentration (see Fig. 3, between 8 and 18 months before the second dose is given).

Finally, the data is longitudinal but the reader is never presented with individual antibody concentration trajectories in the current manuscript. This seems like an ideal place to do this – the

edge effects aside, do the individual Ab concentration trajectories of all the individuals in this cohort look like they're on track to wane by the middle of the third year of life? This child who is seronegative at 26 months and is driving this result – were they actually seropositive after vaccination and waned to susceptibility in that short timeframe, or were they seronegative at their earlier time points as well? If they were seropositive in their earlier samples, was their antibody concentration response high and waned rapidly, or were they just above threshold and waned a little to just below threshold? See for example, the presentation in Figure 4 of Brinkman et al JID 2019;220:594–602 for an example of the sort of presentation I think could be enlightening here. Stable, high levels of immunity from two doses at 8 and 18 months through at least 9 years of age clearly supported by the data presented in Figure 3 A/B

A catchup dose between 8 months and 5 years reduces the relative risk of being susceptible The presentation of this is confusing and seems internally inconsistent. The paragraph at lines 179-195 seems to argue both sides, that there are no significant differences between the 2 and 3 dose cohorts, and that the extra dose provides statistically significant risk reduction. It's hard to see how they can both be true.

My interpretation of Figure 4 is that it says that a child who receives 2 doses of MCV, at 8 and 18 months, faces about a 40% chance of having sero-reverted to susceptibility by age 7 (blue line, group 1)? This seems out of line with the data presented in Figure 3, which shows high levels of antibody and low levels of susceptibility through the 9-year endpoint. Supplemental Figure 5 presents the modeled % susceptible among 2-dose children with the higher 300 mIU/ml threshold, and at age 7 the % susceptible still appears to be maybe 15%, albeit with very wide error bars. So I'm unclear how to square that with the much higher probability of having lost immunity by age 7 after 2 doses being presented in Figure 4.

Lastly, switching to a protective threshold of 300 mIU/ml here because there were too few susceptibles under the 200 mIU/ml threshold used elsewhere in the study seems like moving the goalposts to make the case for a third dose.

Data and Methodology/Analytical Approach

The data itself is rather exciting. A longitudinal antibody concentration dataset, with histories of vaccination taking place over somewhat broad age distributions (S Fig. 2), paired mother/neonate samples, and 3 years worth of followup on a cohort of both neonates and older children offers a unique opportunity for study of the dynamics of vaccine priming and boosting, the individual-level variance in antibody response to vaccination, the dependence of that response on pre-vaccination antibody concentrations, and the durability of immunity. The authors focus in primarily on the durability of immunity question

I am not an expert on the laboratory side, performing the serological assays and generating such a dataset. Looking at presentations of the data like Figure S3B, I don't see any obvious causes for concern.

The results of the GAMM fits are under-reported to the audience. The model is described as including covariates for sex, gestational age, mode of delivery, birth weight, whether the child was breastfed, and a random effect by child. The authors should report whether any of these covariates were statistically significant, what the magnitudes of the effects were. In particular, I would be very interested to know how much of the individual-level variance in vaccine response is explained by these factors vs. having to be accounted for in the random effect.

The choice to fit two separate GAMM models, one to the antibody concentration and one to the proportion susceptible, is a curious choice to me. It admits the possibility for discordant results between the two models, and we do in fact see this upon close inspection of the figures. For example, in Figure 2C, the Ab concentration distribution has apparently reached a high plateau by age 10 or 11 months, but in Figure 2D, the % susceptible at these ages is rapidly declining but still quite high, 25-50% or so. These are two results, from the same dataset, that seemingly can't both be true at the same time, which calls into question the approach. Given that susceptibility is defined relative to an antibody concentration threshold, I think a superior approach would be to model the overall distribution of the antibody concentration over time, rather than just the mean, and then define the proportion susceptible as the the proportion of that distribution below the protective threshold.

The smoothed function of age also seems to allow behavior that is in conflict with general immunological priors – i.e., in the absence of exposure or boosting, antibody levels should stay stable or decline with age, and the proportion susceptible should stay stable or increase with age.

As structured, in e.g., Figure 3D or Supplemental Figure 4, the proportion susceptible wiggles up and down repeatedly in a way that doesn't make much biological sense. It's a little nitpicky, perhaps, as I know this is an artifact of the model structure and these wiggles are within the noise bounds. But it would seem fairly straightforward to enforce, or at least to smooth over longer timescales or impose a larger curvature penalty on the function of age far from a vaccination event (either of those may also strongly impact the model's prediction about waning immunity from a single dose).

Limiting the models to fit only to the cohort of children vaccinated exactly at 8 months, or exactly at 8 to 18 months, simplifies the model but I think leaves interesting science on the table. There are sizable cohorts of children in the dataset who receive MCV1 at 9 and 10 months – do their seroconversion rates and waning profiles differ appreciably from the 8 month children? There are smaller (still 10s of samples) of children who got the first dose at 7 months, or in the 19-22 month age bin, can anything be learned from them? Similarly, there are decent-sized cohorts of children who get MCV2 at 10-16 months, at 19-22 months, and in the 3rd year of life as opposed to strictly at 18 months, and they are left completely unexplored as well. Restricting to these exact months does allow for a very simple model, e.g., just using a smooth function of age to capture the antibody dynamics since all of the vaccinations take place at the same age for all children in the group, but it's also a less satisfying approach that seems to leave a lot on the table. A model that was aware of the timing of a vaccination dose and the pre-vaccination antibody concentrations, and estimated the magnitude and variance of the antibody response to that vaccination, along with a waning model between/after vaccination timepoints, would be a lot more satisfying to me, and would leverage the full dataset rather than restricting to smaller sub-groups.

The Kaplan-Meier analysis of waning to susceptibility appears to be in direct conflict with the GAMM models, showing a high risk of waning to susceptibility by age 7 while the GAMM models show no significant waning by age 9. The authors take no real steps to explain this discrepancy, which is disappointing as it calls into question the basis for the claim about the efficacy of a third dose.

Clarity and Context

Sufficient context about the importance of the questions the authors aim to address here is provided in the introduction and discussion. There are a few points where I think the authors could adjust the manuscript for clarity.

Line 110 – This seems to me a good place to discuss the “two-cohort” nature of the study population. Quantities like the median age at enrollment are not that useful/descriptive of the actual study population of some 40% neonates enrolled at birth, and 60% children ranging from 1-9 years, and so I'd suggest describing that structure in text, rather than reporting the median age, median followup time, etc.

Line 121-123: “Among 1,431 children with complete immunization records, 335 children received MCV1 vaccination at 8 months, and 300 children received two routine MCV doses at ages 8 and 18 months (Supplemental Figure 2)”. In Figure S2, it appears to me that 834 children received MCV1 at 8 months, not 335. More clarity on the selection of the population to be modeled would be useful, here or in the Methods section

Figure 2 – please check the numbers under Figure 2B. It appears to me that “No. susceptible” is really “No. immune”.

Reviewer #2 (Remarks to the Author):

I would like to congratulate the authors on the completion of a very well planned investigation, thorough analysis and well-written manuscript.

I did have a few comments/questions for clarification:

1. Abstract, lines 43: Can I clarify whether the antibody concentration remained above the

protective threshold for the duration of the study, which was 8.5 years? This seems to be implied on line 170 of the manuscript. If that is the case, it seems that the abstract should note that the antibody concentration remained above the protective threshold for the duration of the study or at least 8.5 years rather than stating that it was "for" 8.5 years, which to me implies that it dropped below the protective concentration following that time period.

2. Abstract, sentence starting on line 47: I'm not sure that I agree with the statement that the study findings show that target s could be optimized. I do however, believe that the study findings offer important data points the should be considered by national immunization programs when optimizing the schedule for administration of MCV routine and supplemental doses.

3. Introduction, line 68-69: Minor wording suggestion: I would suggest to insert the word "worldwide" after "born" in line 68 and delete in line 69.

4. Introduction, line 92: I think it should be "immune" rather than "immunity".

5. Introduction, lines 98 and following: I do think readers would be interested in an additional sentence describing why China provides first dose at 8 months and most other countries offer the first dose at 9 months, if that is possible.

6. Introduction, line 103: I wonder if it would be more clear to state "measles antibody kinetics of maternal antibody and following vaccination".

7. Results, lines 188 to 191: The finding that a supplemental dose significantly reduced susceptibility seems to me to be one of the most important findings of the article and should be highlighted in the abstract.

8. Discussion, lines 207-209: I'm confused by this sentence. It is related to comment #1, above. Is it for the duration of the study?

9. Discussion, lines 300-302: I wonder if the authors could be more explicit in what is being suggested. The meaning of the phrase "optimizing the age target" is unclear to me. Should China drop first dose below 8 months? What about the need for a third dose (or supplemental) dose in the first five years?

REVIEWER COMMENTS

Reviewer #1 (Remarks to the Author):

Significance

Overall, the manuscript aims to address an important topic of major public health interest – the timing of routine measles vaccination of infants, in an era and setting where maternally-derived protection is waning faster due to a cohort of mothers whose immunity is largely vaccine-derived rather than from natural infection and who have not faced repeated challenge by the measles virus as they live in a near-elimination setting. This is a critical decision, trading off the risk of measles infection in infancy vs. the potential for suboptimal vaccine response and thus infection later in life, and there remains a good deal of uncertainty about the magnitude and durability of the antibody response to earlier MCV doses. The authors also have a cohort of children who have received 3 measles vaccine doses, which is not recommended in any schedule that I'm aware of, but I at least believe is an open question as to whether a third dose might potentially be necessary to maintaining population immunity over long timescales in elimination settings. The authors have identified a dataset that appears very promising in addressing the uncertainty around these questions.

Response: We would like to thank the reviewer for the generally positive assessment of our manuscript and for acknowledging the importance of this topic. We are also very grateful for the constructive feedback that allowed us to improve our manuscript.

Key Results

The manuscript describes a longitudinal dataset of serum samples, with a cohort of neonates and a cohort of older children 1-10 years of age, followed up over three years. The samples were tested for measles IgG antibody, and on the basis of this data the manuscript makes four central claims:

1. Waning of maternally-derived immunity with a half-life of about 2 months
2. Immunity from a single dose of MCV wanes by 26 months of age
3. Two MCV doses at 8 and 18 months provide high levels of population immunity, stable through at least 9 years of age
4. A catchup dose between 8 months and 5 years reduces a child's relative risk of becoming susceptible (clarity needed in the statement of this claim – becoming susceptible by what age?)

Response: We would like to thank the reviewer for the in-depth assessment of our manuscript.

As suggested, in the revised manuscript, we have now specified the age when describing and analysing the effects of a catch-up dose in addition to the routine cycle. We also agree with the reviewer that the wording “risk of becoming susceptible” is rather unclear and misleading and in the current version of the manuscript we specified the epidemiological outcomes obtained with Kaplan-Meier methods (*i.e.*, the cumulative incidence of seroreversion).

In the Abstract:

“A catch-up MCV dose in addition to the routine cycle between 8 months and 5 years reduced the cumulative incidence of seroreversion by 79.3-88.7% by the age of 6 years.”

In the Results section:

“In particular, by the age of 6 years, a catch-up dose between 8 and 18 months of age showed a 79.3% (95%CI 70.5-100%) reduction in the cumulative incidence of seroreversion, and a catch-up dose between 2 and 5 years of age was associated with an 88.7% (95%CI 83.5-100%) reduction by assuming a protective threshold of 200 mIU/ml (Fig. 4).”

Validity

I find the basis for two of these key results very unconvincing in the current analysis. here are also a number of statements that appear to be internally inconsistent

Waning of maternally-derived immunity with a half-life of about 2 months: very clearly supported by the data in Figs. 2A/B

Response: We thank the reviewer for the positive assessment of our findings regarding the early waning of maternally derived immunity.

Waning of single-dose immunity by 26 months of age: Very unconvincing. Looking at Figures 2A and B, the data presented here seem to me to provide thin support for making such a strong claim about antibody waning. The finding appears to be driven by a single seronegative, out of three samples, in the final age bin at 26-27 months (in discussion, at lines 246-247, the authors say it is driven by three individuals who were categorized as seronegative at 26-27 months, but Figure 2B indicates that there are 3 total individuals at 26-27 months, only 1 of whom is seronegative).

To me, this seems like an artifact of a model given too much freedom to curve sharply to match a single data point at the edge of the fit range (Fig. 2D). I’m curious what the authors would find if they re-fit the % susceptible model under repeated bootstrapping of the data; I would guess that, given the

lack of any susceptibles found in the 18-23 month age bins, the duration of protection bounces from 26 months to lifelong based on whether that 1 seronegative is in the bootstrap sample or not.

There are other causes for concern about this result. The model output presented in Fig 2D has unrealistically low variance on the proportion susceptible in the 15-22 month age bin given the error bars on the data shown in Fig. 2B; the model really overstates confidence here. And there are distinctly internally inconsistent statements being made in this part of the results – at line 157: “susceptibility to infection decreased ... to 1.1% at 24 months”, but at 159 “seroconversion rate following MCV1 at 8 months is estimated to be 89.6%”. That’s a full order of magnitude difference in effectiveness. Similarly, 1.1% susceptibility at 24 months is quite hard to square with a duration of protection of only 26 months (with a 95% interval down to 23.6 months); this would require remarkably rapid and synchronous waning across a population showing a lot of variance in Ab concentration (see Fig. 3, between 8 and 18 months before the second dose is given).

Finally, the data is longitudinal but the reader is never presented with individual antibody concentration trajectories in the current manuscript. This seems like an ideal place to do this – the edge effects aside, do the individual Ab concentration trajectories of all the individuals in this cohort look like they’re on track to wane by the middle of the third year of life? This child who is seronegative at 26 months and is driving this result – were they actually seropositive after vaccination and waned to susceptibility in that short timeframe, or were they seronegative at their earlier time points as well? If they were seropositive in their earlier samples, was their antibody concentration response high and waned rapidly, or were they just above threshold and waned a little to just below threshold? See for example, the presentation in Figure 4 of Brinkman et al JID 2019;220:594–602 for an example of the sort of presentation I think could be enlightening here.

Response: We thank the reviewer for raising these important points about the waning of immunity following MCV1 vaccination at 8 months.

As suggested, we have included individual antibody concentration trajectories in the Appendix (Fig. S4), and revised the number of seronegative individuals at 26-27 months (n=1) in the Discussion section. We have also added a new analysis to understand how the seronegative child at 26-27 months affected the estimated persistence of MCV1-induced antibodies, in which the dynamics of IgG antibody concentrations were refitted using repeatedly drawn bootstrap samples from all observations or from observations that excluded the seronegative child at 26-27 months. As correctly guessed by the reviewer, the fitted antibody response and persistence after MCV1 vaccination at 8 months were significantly affected by this seronegative child at 26-27 months. This has been clarified in the main text and included in the Appendix (Fig. S6).

“In addition, in sensitivity analyses, we considered the uncertainties in the estimated

persistence of MCV1-induced antibodies due to individual heterogeneities in the immune response to MCV1 at age 8 months. In such analyses, the immune response to MCV1 was fitted using repeatedly drawn bootstrap samples from all observations or from observations that excluded the seronegative child at 26-27 months (Supplementary Information, p18)."

In addition, to provide more robust estimates and avoid potential bias, we made the following revisions. **First**, we removed a set of observations from the main analysis when characterizing the immune responses prior to and following MCV1 vaccination. Specifically, we excluded three observations at ages 26-27 months (including one seronegative individual) and reported only the dynamics of the IgG antibody response to MCV1 at 8 months until 23 months of age, because of their significant impact on the robustness of the estimated persistence of antibodies after MCV1 vaccination at 8 months (described above). While calculating the seroconversion rate after MCV1 vaccination at 8 months of age, we additionally excluded four children who were previously misclassified as individuals with paired serum samples between 6 and 24 months of age. **Second**, we have now adopted a new methodological framework (generalized additive model, GAM) to determine the immune response against measles prior to and following different vaccination schedules. In such models, we considered the magnitude and variance of the antibody response across ages and by children's vaccination status (including pre-vaccination, after a single dose vaccination at 8 months, or after two-dose vaccination at 8 and 18 months). Using these GAM models, we found that the fitted dynamics of IgG antibody concentration and of the fraction of susceptible individuals, and immune protection persistence after MCV1 at 8 months were more robust (see updated Fig. 2, Fig. 3, Fig. S5 and Fig. S8). It is estimated that IgG antibody levels at 53 months post MCV1 vaccination drop below the protective threshold of 200 mIU/ml, which was consistent with Brinkman et al.'s recent study. **Third**, we recalculated the seroconversion rate after MCV1 vaccination at 8 months. In the revised manuscript, the seroconversion rate is estimated to be 98.6% (95% CI 92.2-99.9%), which is now consistent with updated estimates of the fraction of seropositive individuals at 24 months (98.2%, 95% CI 91.1-100%).

The text has been revised in depth as follows:

In the Methods section:

"Incorporating children's age and vaccination status (including pre-vaccination, after a single dose vaccination at 8 months, or after two-dose vaccination at 8 and 18 months), we then used generalized additive models (GAMs) to fit the log of IgG antibody concentrations and the fraction of susceptible individuals at different ages

among children with either one dose of MCV1 or the routine two-dose MCV (Supplementary Information, p7)."

In the Results section:

"Since we had only three observations for children between 26 and 28 months, including one seronegative result that can affect the estimated kinetics of the immune response to MCV1 at 8 months, in the main analysis, we reported MCV1-induced antibody response and persistence until 23 months of age. The Appendix shows a sensitivity analysis where these observations are not excluded (Supplementary Figs. 4 and 6)."

"The fraction of susceptible individuals decreased sharply from 98.8% (95%CI 97.9-99.8%) at 8 months, to 5.0% (95%CI 0-10.4%) at 9 months, and to 1.8% (95%CI 0-8.9%) at 24 months (Fig. 2B and 2D). [...] By extrapolating generalized additive model (GAM) fitted parameter values beyond the observed 24 months of age, it was estimated that IgG antibody concentrations would drop below the protective threshold of 200 mIU/ml at 53 months post MCV1 administration (i.e., 5.1 years of age) (Supplementary Fig. 7A)."

"[...] Using paired serum samples from infants at 6 and 24 months of age (n=69), the proportion of children with seroconversion or a fourfold rise in IgG antibody following MCV1 at 8 months was estimated to be 98.6% (95%CI 92.2-99.9%)."

In the Discussion section:

"Following MCV1 administration at 8 months, extrapolated IgG antibody concentration levels remained above a protective threshold of 200 mIU/ml until 53.0 months post MCV1, suggesting a target age for MCV2 at approximately 4 years of age. This finding was consistent with Brinkman et al.'s recent study, in which they demonstrated that neutralizing antibody levels dropped below the cut-off for clinical protection at 60 months post MCV1 administration at age 6-8 months²⁴. Although a 53.0-month antibody persistence after MCV1 at 8 months was much shorter than that reported in a systematic review of MCV persistence (approximately 35 years post-MCV1)²⁸, the difference could be explained by a different measles infection risk at the time of our study with respect to other studies included in the meta-analysis. In the Appendix we show that the duration of protection following MCV1 vaccination is sensitive to three observations at 26-27 months, which we excluded from the main analysis due to a lack of information about their status after MCV1 vaccination.

Further modeling work is needed to reconstruct individual immune response profiles against measles following different combinations of MCV1 and MCV2 schedules to determine context-specific optimal schedules for MCV vaccination.”

We would like to express once again our gratitude to the reviewer for raising these points, as we believe our analysis has remarkably improved thanks to them.

Stable, high levels of immunity from two doses at 8 and 18 months through at least 9 years of age clearly supported by the data presented in Figure 3 A/B

Response: Thank you again for the positive evaluation of our findings.

A catchup dose between 8 months and 5 years reduces the relative risk of being susceptible The presentation of this is confusing and seems internally inconsistent. The paragraph at lines 179-195 seems to argue both sides, that there are no significant differences between the 2 and 3 dose cohorts, and that the extra dose provides statistically significant risk reduction. It’s hard to see how they can both be true.

My interpretation of Figure 4 is that it says that a child who receives 2 doses of MCV, at 8 and 18 months, faces about a 40% chance of having sero-reverted to susceptibility by age 7 (blue line, group 1)? This seems out of line with the data presented in Figure 3, which shows high levels of antibody and low levels of susceptibility through the 9-year endpoint. Supplemental Figure 5 presents the modeled % susceptible among 2-dose children with the higher 300 mIU/ml threshold, and at age 7 the % susceptible still appears to be maybe 15%, albeit with very wide error bars. So I’m unclear how to square that with the much higher probability of having lost immunity by age 7 after 2 doses being presented in Figure 4.

Lastly, switching to a protective threshold of 300 mIU/ml here because there were too few susceptibles under the 200 mIU/ml threshold used elsewhere in the study seems like moving the goalposts to make the case for a third dose.

Response: We would like to thank the reviewer who helped us realize that the wording “risk of being susceptible” used in the original manuscript was very unclear and misleading. In this study, when characterizing susceptibility profiles prior to and following different vaccination schedules, we used two different epidemiological outcome measurements: the fraction of susceptible individuals and the cumulative incidence of seroreversion. The fraction of susceptible individuals and the cumulative incidence of seroreversions were obtained with generalized additive models and Kaplan-Meier methods, respectively. In particular, the former quantified the proportion of susceptible individuals at a given age, whereas the latter measured the cumulative occurrence of seroreversions by a given age. As such, for individuals within the

same vaccination group, we could observe that their cumulative incidence of seroreversion (Fig. 4) was much higher than the fraction of susceptible individuals (e.g., Fig. 3 and Fig. S5) at a given age. We have revised the text in the Methods section to clarify this:

“[...] we then used generalized additive models (GAMs) to fit the log of IgG antibody concentrations and the fraction of susceptible individuals at different ages among children with either one dose of MCV1 or the routine two-dose MCV (Supplementary Information, p7).”

“[...] Kaplan–Meier survival methods were used to calculate the cumulative incidence of seroreversion in IgG antibodies by a given age.”

To improve clarity regarding the effects of a catch-up dose in addition to the routine cycle, we have also fully rewritten the text in the Results and Discussion sections:

“Age-specific IgG GMCs and the fraction of susceptible individuals among children with and without a catch-up dose in addition to the routine cycle (i.e., the routine and catch-up vaccination groups) are shown in Supplementary Fig. 9. Although IgG GMCs declined with age in both vaccination groups, IgG GMCs were generally higher in the catch-up vaccination group than in the routine vaccination group within the same age group, regardless of age at catch-up vaccination (Supplementary Fig. 9A-9B). A continuous increase in the fraction of susceptible individuals with age was seen in the routine vaccination group, whereas the increase was not obvious in the catch-up vaccination group (Supplementary Fig. 9C-9D). We also found that, since the third year of the child’s life, children in the routine vaccination group had a significantly higher cumulative incidence of seroreversion than those in the catch-up vaccination group, regardless of which protective threshold was used (p -value <0.001 , Fig. 4, Supplementary Fig. S10). In particular, by the age of 6 years, a catch-up dose between 8 and 18 months of age showed a 79.3% (95%CI 70.5-100%) reduction in the cumulative incidence of seroreversion, and a catch-up dose between 2 and 5 years of age was associated with an 88.7% (95%CI 83.5-100%) reduction by assuming a protective threshold of 200 mIU/ml (Fig. 4).”

“In addition, we observed that a catch-up dose in addition to the routine cycle between ages 8 months and 5 years could delay the decay of IgG antibody concentrations and increases of susceptibility at the population level, and have substantially reduced individuals’ cumulative incidence of seroreversion before 6 years of age, regardless of

age at catch-up vaccination.”

Moreover, we agree that assuming a protective threshold of 300 mIU/ml in the analyses may seem arbitrary, and at the same time it may inflate the effects of a catch-up dose in addition to the routine cycle. Therefore, in the revised manuscript, the main analysis uses a protective threshold of 200 mIU/ml to quantify the contribution of a catch-up dose in addition to the routine cycle to the cumulative incidence of seroreversion by age 6 years, whereas an alternatively protective threshold of 300 mIU/ml was used in a sensitivity analysis presented in the Appendix (Fig. S10). We found that selecting protective thresholds did not affect the estimated reduction in the cumulative incidence of seroreversion due to a catch-up dose in addition to the routine cycle. This has now been clarified in the main text.

“For the Kaplan–Meier analyses, we used a threshold of 200 mIU/ml to define seroreverted individuals in the main analyses, whereas 300 mIU/ml was selected as an alternative threshold in the sensitivity analysis.”

“We also found that, since the third year of the child’s life, children in the routine vaccination group had a significantly higher cumulative incidence of seroreversion than those in the catch-up vaccination group, regardless of which protective threshold was used (p -value <0.001 , Fig. 4, Supplementary Fig. S10).”

Data and Methodology/Analytical Approach

The data itself is rather exciting. A longitudinal antibody concentration dataset, with histories of vaccination taking place over somewhat broad age distributions (S Fig. 2), paired mother/neonate samples, and 3 years worth of followup on a cohort of both neonates and older children offers a unique opportunity for study of the dynamics of vaccine priming and boosting, the individual-level variance in antibody response to vaccination, the dependence of that response on pre-vaccination antibody concentrations, and the durability of immunity. The authors focus in primarily on the durability of immunity question

I am not an expert on the laboratory side, performing the serological assays and generating such a dataset. Looking at presentations of the data like Figure S3B, I don’t see any obvious causes for concern.

Response: We would like to thank the reviewer for taking the time to assess our study.

The results of the GAMM fits are under-reported to the audience. The model is described as including covariates for sex, gestational age, mode of delivery, birth weight, whether the child was breastfed, and a random effect by child. The authors should report whether any of these covariates were

statistically significant, what the magnitudes of the effects were. In particular, I would be very interested to know how much of the individual-level variance in vaccine response is explained by these factors vs. having to be accounted for in the random effect.

Response: We apologize for the lack of discussion. As suggested, we have now built a generalized linear mixed model (GLMM) to understand how much of the variance in antibody response can be explained by children's baseline characteristics, age, vaccination status and their individual heterogeneities. In particular, we have added two new tables to the Appendix (Tab. S2-3). We found that none of the children's baseline characteristics were associated with IgG antibodies, and there were very few heterogeneities in individuals' immune response against measles. As such, we have now included age and vaccination status only in the new GAM, which are significantly associated with IgG antibodies. We have also expanded the text in the Methods and Results section to comment on this:

“A generalized linear mixed-effects model (GLMM) was built to quantify the effects of potential factors associated with IgG antibodies (Supplementary Information, p6). Incorporating children's age and vaccination status (including pre-vaccination, after a single dose vaccination at 8 months, or after two-dose vaccination at 8 and 18 months), we then used generalized additive models (GAMs) to fit the log of IgG antibody concentrations and the fraction of susceptible individuals at different ages among children with either one dose of MCV1 or the routine two-dose MCV (Supplementary Information, p7).”

“It is important to identify factors associated with children's measles-specific IgG antibodies before fitting the kinetic curves of antibody concentrations and of the fraction of susceptible individuals by age. We thus performed a multivariate regression analysis with a generalized linear mixed model (GLMM), which found only two statistically significant factors associated with measles-specific IgG antibodies in children: age and the number of MCV doses (MCV1 vs. non-vaccination: p -value <0.001 ; MCV2 vs. non-vaccination: p -value <0.001) (Supplementary Table 2). In particular, the IgG antibody concentration decreased with age (p -value <0.001). We found that children's baseline characteristics, age and vaccination status together accounted for 32.3% of the total variance (Supplementary Table 3). Furthermore, individual heterogeneities in measles-specific immune response were low, accounting for only 7.8% of the total variance.”

The choice to fit two separate GAMM models, one to the antibody concentration and one to the proportion susceptible, is a curious choice to me. It admits the possibility for discordant results

between the two models, and we do in fact see this upon close inspection of the figures. For example, in Figure 2C, the Ab concentration distribution has apparently reached a high plateau by age 10 or 11 months, but in Figure 2D, the % susceptible at these ages is rapidly declining but still quite high, 25-50% or so. These are two results, from the same dataset, that seemingly can't both be true at the same time, which calls into question the approach. Given that susceptibility is defined relative to an antibody concentration threshold, I think a superior approach would be to model the overall distribution of the antibody concentration over time, rather than just the mean, and then define the proportion susceptible as the proportion of that distribution below the protective threshold.

Response: We would like to thank the reviewer for raising this very important issue. The inconsistencies between the fitted IgG antibody concentration and fraction of susceptible individuals were the result of inappropriate smoothing parameters in the original GAMM models. In the revised manuscript we adopted a new methodological framework (GAM) that consistently modelled the decreases in IgG antibody concentration and the corresponding increases in the fraction of susceptible individuals (Fig. 2).

Moreover, as properly pointed out by the reviewer, it is true that we can use an individual model to fit each child's antibody concentration over time and then determine population susceptibility in accordance with the distribution of fitted individual antibody concentrations. However, projecting individual-specific IgG antibody profiles would require the design and development of a different approach, which we believe falls outside the scope of this study. As such, we have added the following sentence to acknowledge this study limitation:

“Fifth, the GAM model used in this study tended to smooth observations in each age group, and failed to account for individual antibody concentration trajectories, particularly for those who had either a low response or no response to MCV. This might directly lead to an underestimated IgG antibody concentration and persistence following different vaccination schedules. More individuals with consecutive serum samples before and after each MCV vaccination should be involved to reconstruct individual immune response profiles against measles prior to and following different vaccination schedules.”

The smoothed function of age also seems to allow behavior that is in conflict with general immunological priors – i.e., in the absence of exposure or boosting, antibody levels should stay stable or decline with age, and the proportion susceptible should stay stable or increase with age. As structured, in e.g., Figure 3D or Supplemental Figure 4, the proportion susceptible wiggles up and down repeatedly in a way that doesn't make much biological sense. It's a little nitpicky, perhaps, as I know this is an artifact of the model structure and these wiggles are within the noise bounds. But it

would seem fairly straightforward to enforce, or at least to smooth over longer timescales or impose a larger curvature penalty on the function of age far from a vaccination event (either of those may also strongly impact the model's prediction about waning immunity from a single dose).

Response: We would like to thank the reviewer for pointing this out. As mentioned above, we are now using GAM models (instead of GAMM models) to fit the immune response against measles prior to and following different vaccination schedules. In the revised manuscript, the estimated IgG antibody concentration and fraction of susceptible individuals steadily decreased or increased with increasing age (see updated Fig. 2, Fig. 3 and Fig. S5).

Limiting the models to fit only to the cohort of children vaccinated exactly at 8 months, or exactly at 8 to 18 months, simplifies the model but I think leaves interesting science on the table. There are sizable cohorts of children in the dataset who receive MCV1 at 9 and 10 months – do their seroconversion rates and waning profiles differ appreciably from the 8 month children? There are smaller (still 10s of samples) of children who got the first dose at 7 months, or in the 19-22 month age bin, can anything be learned from them? Similarly, there are decent-sized cohorts of children who get MCV2 at 10-16 months, at 19-22 months, and in the 3rd year of life as opposed to strictly at 18 months, and they are left completely unexplored as well. Restricting to these exact months does allow for a very simple model, e.g., just using a smooth function of age to capture the antibody dynamics since all of the vaccinations take place at the same age for all children in the group, but it's also a less satisfying approach that seems to leave a lot on the table. A model that was aware of the timing of a vaccination dose and the pre-vaccination antibody concentrations, and estimated the magnitude and variance of the antibody response to that vaccination, along with a waning model between/after vaccination timepoints, would be a lot more satisfying to me, and would leverage the full dataset rather than restricting to smaller sub-groups.

Response: Thank you for the kind suggestion. We fully agree with the reviewer about the importance of investigating the seroconversion rates and waning profiles among children who received MCV1/MCV2 at different ages by use of our full dataset. In fact,

[REDACTED]

[REDACTED]

Moreover, we have largely expanded the Discussion section to acknowledge our limitations regarding the ability to shed light on the differences in vaccine-induced immune responses in sizable cohorts of children with MCV1 and MCV2 vaccination at different ages in this study:

“Additionally, restricting the study to participants who had received 2 doses of measles vaccine at ages consistent with the NIP recommendation did not allow us to shed light on the differences in vaccine-induced immune responses in sizable cohorts of children with MCV1 and MCV2 vaccination at different ages. Further studies are needed to capture the antibody dynamics following different combinations of MCV1 and MCV2 vaccination schedules.”

The Kaplan-Meier analysis of waning to susceptibility appears to be in direct conflict with the GAMM

models, showing a high risk of waning to susceptibility by age 7 while the GAMM models show no significant waning by age 9. The authors take no real steps to explain this discrepancy, which is disappointing as it calls into question the basis for the claim about the efficacy of a third dose.

Response: We apologize for the ambiguous terms used in the original manuscript. As mentioned above, we have now extensively revised the text to clearly define the outcomes obtained with GAM and Kaplan-Meier analyses:

“[...] we then used generalized additive models (GAMs) to fit the log of IgG antibody concentrations and the fraction of susceptible individuals at different ages among children with either one dose of MCV1 or the routine two-dose MCV (Supplementary Information, p7).”

“[...] Kaplan–Meier survival methods were used to calculate the cumulative incidence of seroreversion in IgG antibodies by a given age.”

Moreover, we have fully rewritten the following paragraph to show the effects of a catch-up dose in addition to the routine cycle:

“Age-specific IgG GMCs and the fraction of susceptible individuals among children with and without a catch-up dose in addition to the routine cycle (i.e., the routine and catch-up vaccination groups) are shown in Supplementary Fig. 9. Although IgG GMCs declined with age in both vaccination groups, IgG GMCs were generally higher in the catch-up vaccination group than in the routine vaccination group within the same age group, regardless of age at catch-up vaccination (Supplementary Fig. 9A-9B). A continuous increase in the fraction of susceptible individuals with age was seen in the routine vaccination group, whereas the increase was not obvious in the catch-up vaccination group (Supplementary Fig. 9C-9D). We also found that, since the third year of the child’s life, children in the routine vaccination group had a significantly higher cumulative incidence of seroreversion than those in the catch-up vaccination group, regardless of which protective threshold was used (p-value<0.001, Fig. 4, Supplementary Fig. S10). In particular, by the age of 6 years, a catch-up dose between 8 and 18 months of age showed a 79.3% (95%CI 70.5-100%) reduction in the cumulative incidence of seroreversion, and a catch-up dose between 2 and 5 years of age was associated with an 88.7% (95%CI 83.5-100%) reduction by assuming a protective threshold of 200 mIU/ml (Fig. 4).”

“In addition, we observed that a catch-up dose in addition to the routine cycle between ages 8 months and 5 years could delay the decay of IgG antibody concentrations and increases of susceptibility at the population level, and have substantially reduced individuals’ cumulative incidence of seroreversion before 6 years of age, regardless of age at catch-up vaccination.”

Clarity and Context

Sufficient context about the importance of the questions the authors aim to address here is provided in the introduction and discussion. There are a few points where I think the authors could adjust the manuscript for clarity.

Line 110 – This seems to me a good place to discuss the “two-cohort” nature of the study population. Quantities like the median age at enrollment are not that useful/descriptive of the actual study population of some 40% neonates enrolled at birth, and 60% children ranging from 1-9 years, and so I’d suggest describing that structure in text, rather than reporting the median age, median followup time, etc.

Response: Thank you for the suggestion. We have added the following text to the Results section and specified the sample size for each enrolment date.

“We enrolled 2,629 children in this study (Fig. 1); 555, 352, 354, 318, 335, 316, and 399 children were enrolled at birth and at 1, 2, 3, 4, 5, and 6-9 years of age, respectively.”

Line 121-123: “Among 1,431 children with complete immunization records, 335 children received MCV1 vaccination at 8 months, and 300 children received two routine MCV doses at ages 8 and 18 months (Supplemental Figure 2)”. In Figure S2, it appears to me that 834 children received MCV1 at 8 months, not 335. More clarity on the selection of the population to be modeled would be useful, here or in the Methods section

Response: We thank the reviewer for pointing this out; there was indeed a mistake in the sentence. The sentence now reads “[...] 834 (58.3%) children received MCV1 at 8 months (Supplementary Fig. 2), and 300 (21.0%) children received two routine doses at ages 8 and 18 months.”

Figure 2 – please check the numbers under Figure 2B. It appears to me that “No. susceptible” is really “No. immune”.

Response: Thank you; the correction has been made.

Reviewer #2 (Remarks to the Author):

I would like to congratulate the authors on the completion of a very well planned investigation, thorough analysis and well-written manuscript.

Response: We would like to thank the reviewer for the positive assessment of our manuscript and the useful comments.

I did have a few comments/questions for clarification:

1. Abstract, lines 43: Can I clarify whether the antibody concentration remained above the protective threshold for the duration of the study, which was 8.5 years? This seems to be implied on line 170 of the manuscript. If that is the case, it seems that the abstract should note that the antibody concentration remained above the protective threshold for the duration of the study or at least 8.5 years rather than stating that it was "for" 8.5 years, which to me implies that it dropped below the protective concentration following that time period.

Response: We apologize for the lack of clarity. In light of this and other comments by Reviewer #1, we have added a new analysis to estimate the immune protection duration after two routine MCV doses at 8 and 18 months of age (Fig. S7). We have thus revised the text accordingly:

In the Abstract section:

“Following a second measles-containing vaccine dose at 18 months of age, we that estimate antibody concentrations fall below the protective threshold at age 20.6 years.”

In the Results section:

“Moreover, the simulated evolution of IgG antibody concentrations showed that, 19.1 years after MCV2 vaccination (i.e., 20.6 years of age), IgG antibodies would drop below the protective threshold (Supplementary Fig. 7B).”

In the Discussion section:

“In a population of children who received 2 doses of measles vaccine at ages consistent with the recommendation of China's national immunization program (NIP) (i.e., 8 and 18 months), the IgG antibody concentration fell below protective levels at 20.6 years of

age.”

2. Abstract, sentence starting on line 47: I'm not sure that I agree with the statement that the study findings show that target s could be optimized. I do however, believe that the study findings offer important data points the should be considered by national immunization programs when optimizing the schedule for administration of MCV routine and supplemental doses.

Response: Thank you for pointing this out and we agree with the interpretation given by the reviewer. As such, we revised the sentences as follows:

“Our findings show that MCV1 at 8 months induces a good immune response, whereas immune protection after two routine doses at 8 and 18 months is not lifelong. These findings, coupled with the effectiveness of a catch-up dose in addition to the routine cycle, could be instrumental to relevant stakeholders when planning routine immunization schedules and supplemental immunization activities.”

3. Introduction, line 68-69: Minor wording suggestion: I would suggest to insert the word "worldwide" after "born" in line 68 and delete in line 69.

4. Introduction, line 92: I think it should be "immune" rather than "immunity".

Response: Thank you, the corrections has been made.

5. Introduction, lines 98 and following: I do think readers would be interested in an additional sentence describing why China provides first dose at 8 months and most other countries offer the first dose at 9 months, if that is possible.

Response: Thank you for the suggestion. A description of the reason why China provides MCV1 at 8 months has been added:

“[...] , which was developed in accordance with infant immune system development, the age distribution of measles cases, and the levels of maternal and vaccine-induced immunity against measles at a population level ¹⁵.”

6. Introduction, line 103: I wonder if it would be more clear to state "measles antibody kinetics of maternal antibody and following vaccination".

Response: Thank you for pointing this out. The sentence now reads as follows:

“Representative longitudinal serological surveys conducted among Chinese children allow us to simultaneously evaluate the kinetics of measles maternal antibodies and antibodies following vaccination. [...]”

7. Results, lines 188 to 191: The finding that a supplemental dose significantly reduced susceptibility seems to me to be one of the most important findings of the article and should be highlighted in the abstract.

Response: We would like to thank the reviewer for this useful suggestion. We have revised the Abstract to highlight findings regarding the effectiveness of a catch-up dose in addition to the routine cycle:

“A catch-up MCV dose in addition to the routine cycle between 8 months and 5 years reduced the cumulative incidence of seroreversion by 79.3-88.7% by the age of 6 years. [...] These findings, coupled with the effectiveness of a catch-up dose in addition to the routine cycle, could be instrumental to relevant stakeholders when planning routine immunization schedules and supplemental immunization activities.”

8. Discussion, lines 207-209: I'm confused by this sentence. It is related to comment #1, above. Is it for the duration of the study?

Response: We apologize for the lack of clarity. The duration of protection immunity following two routine doses at 8 and 18 months has been clarified in the revised manuscript. The sentence now reads

“[...] the IgG antibody concentration fell below protective levels at 20.6 years of age.”

9. Discussion, lines 300-302: I wonder if the authors could be more explicit in what is being suggested. The meaning of the phrase "optimizing the age target" is unclear to me. Should China drop first dose below 8 months? What about the need for a third dose (or supplemental) dose in the first five years?

Response: Thank you for the suggestion. We have now extensively expanded the text to comment on this:

“In conclusion, our findings suggest that targeting individuals under 8 months of age for MCV1 is necessary to reduce the number of susceptible Chinese children below the age of the currently scheduled vaccination.”

“The estimated reduction in the cumulative incidence of seroreversion due to a catch-up

dose in addition to the routine cycle suggests the effectiveness of nonselective supplementary immunization activities against measles implemented in China, which can be considered a prioritized intervention for reducing both the number of susceptible individuals and of infections in children younger than 5 years of age.”

REVIEWER COMMENTS

Reviewer #1 (Remarks to the Author):

I thank the authors for their thorough consideration and responses to my first round of remarks. I'll focus these comments around the authors' addressing of my two main points of contention in the first round.

1. Concerning the timescale of waning immunity from a single vaccine dose at 8 months, the authors have addressed this via a sensitivity analysis in the supplement and by changing the age endpoint of the main analysis. The sensitivity analysis does demonstrate that the result was highly sensitive to the inclusion of the one seronegative at 28 months of age, and so this result does look more robust by removing them from the sample.

I also thank the authors for adding Figure S4, showing the reader the individual antibody vs. age distributions. At risk of moving the goalposts in a second review, though, this figure reveals a deeper issue - looking at the individual trajectories in Figure S4 reveals that the seronegative child at 28 months who so strongly impacted the waning timescale had apparently never seroconverted to the vaccine in the first place. Removing that child from the analysis of vaccine-derived immunity waning is therefore the right thing to do, as they never had vaccine-derived immunity to wane. This raises the new question for me of whether other non-respondents are present in the sample, and indeed Figure S4 shows that there are. The authors are clear in the text that not every child responds to the vaccine, as expected, but it appears that the children who fail to respond are still being included in the GAM fits that are used to project a timescale for waning of vaccine derived immunity. Failure to seroconvert is a completely separate dynamic from seroconversion followed by waning, and if the goal is to estimate the timescale of waning vaccine immunity, it is absolutely necessary to remove the samples representing children who failed to seroconvert in the first place.

2. My second primary point of contention had to do with perceived disagreement between the "proportion susceptible at age A" in Figure 3 and the "cumulative incidence of seroreversion" in Figure 4. The authors' respond: In this study, when characterizing susceptibility profiles prior to and following different vaccination schedules, we used two different epidemiological outcome measurements: the fraction of susceptible individuals and the cumulative incidence of seroreversion. The fraction of susceptible individuals and the cumulative incidence of seroreversions were obtained with generalized additive models and Kaplan-Meier methods, respectively. In particular, the former quantified the proportion of susceptible individuals at a given age, whereas the latter measured the cumulative occurrence of seroreversions by a given age. As such, for individuals within the same vaccination group, we could observe that their cumulative incidence of seroreversion (Fig. 4) was much higher than the fraction of susceptible individuals (e.g., Fig. 3 and Fig. S5) at a given age.

I am unconvinced that the authors' separation of "proportion susceptible at age A" and "cumulative incidence of seroreversion by age A" resolves this discrepancy. While they are two different quantities, they are not completely independent, they are inextricably linked in a way that this explanation does not resolve - the fraction of susceptible individuals at age A cannot be lower than the cumulative incidence of seroreversion under the conditions here.

More precisely - the fraction of children susceptible at age A is equal to the fraction of children unvaccinated by age A, plus the fraction of children who were vaccinated but failed to seroconvert, plus the fraction of children who seroconverted but seroreverted by age A, minus the fraction of children who still have their maternal antibodies at age A, minus the fraction of children who got measles before age A. The latter two terms are negligible in this setting, and the first term is zero as the sample only includes vaccinated children. I suppose this assumes that nobody suddenly seroconverts to a vaccine dose received in the distant past, but I think that's a reasonable assumption. So the fraction of children susceptible at age A should be equal to the fraction of non-responders + the fraction of responders who serorevert by age A, and so the cumulative incidence of seroreversion by age A cannot be higher than the fraction of susceptible children at age A.

Having read it a couple of times now, my guess at the root of the issue is that the GAM analysis is failing to address the inherent directional correlation that is induced by using age as an independent variable, and the artifacts that appear because of the censored nature of the data (while the Kaplan-Meier statistic is sort of specifically designed to handle those particular features). Instead, the way the data is aggregated to fit the GAM ignores the fact that the data are repeated measurements on individuals, and that age induces one-way correlations (I think this is less of a

problem for the log concentration than for the fraction susceptible). To get concrete – the last 3 bins in Figure 2B contain no susceptibles in the children sampled at those specific ages (18-23 months), so the “Percent susceptible” in each of these these age bins is 0. However, there are 2 susceptibles in the age 16-17 month bin, and 6 in the 14-15 month age bin. Those 8 susceptible kids will still be susceptible when they get to 18-23 months, they just weren’t followed up at those time points. The analysis throws out the information value of that natural history picture, and it makes the GAM results unrealistic –Figure 2D (and 3D as well, if you zoom in) predict that the percent susceptible declines as you age further from the vaccination event, which doesn’t make sense in the absence of measles transmission.

3. I’m interested in the authors’ choice to drop the individual-level random effect when moving from the GLMM model to the GAM model. The justification appears to be in the statement at 154-156: “Furthermore, individual heterogeneities in measles-specific immune response were low, accounting for only 7.8% of the total variance”. I would argue that the entire model only explained 40% of the variance, so the individual heterogeneity accounts for 20% of the model’s explanatory power, hardly negligible.

4. I would appreciate if the authors would please clarify the ethical concerns that prevent making the disaggregated data publicly available. The aggregated data made available in Github (as a side note, the link to the Github repo) appear to only allow someone to plot Figs 2A, 3A, and 4; it does not allow for any meaningful check on the analysis presented herein or to subject the data to further scrutiny. These data are being used in the manuscript to argue for changes to vaccination schedules, a topic with global implications for health, and it seems to me that the components of the raw data critical to that discussion can be provided in a way that protects patient privacy - e.g., removing patient identifying information and demographic characteristics but still providing the disaggregated IgG titer measurements and months at which they were taken, with individual linkage made possible by an anonymous ID. Minimally, what a reader would need to replicate the Supplementary Figure 4; ideally, also including the trajectories for children not shown in Figure S4 because they were vaccinated at non-standard ages.

Reviewer #2 (Remarks to the Author):

I would like to congratulate the authors on the revised manuscript. I believe that the changes made more than adequately address all of the comments provided. There were a few minor grammatical edits that I noted:

1. Abstract line 43: I believe the word "that" at the end of the line is not needed.
2. Introduction line 73: Would it be more clear if you used the phrase "due to" or "caused by" rather than "given an"?
3. line 81: there's a long phrase but I think it should be the "recommended age...has" instead of "have".
4. line 91: Should "and interfere" be changed to "without interfering"?
5. lines 114, 116, 247, and 364: I would suggest to use the word "doses" rather than "cycle".
6. line 208 consider "following MCV2" rather than "of MCV2"
7. lines 248-249: I would suggest to re-phrase "concentrations and increases of" to "concentrations which leads to increase in susceptibility", which I believe is more clear.
8. line 272: Consider changing "things that need to be considered" to "considerations"

Otherwise, the manuscript looks great. The analysis looks great and the conclusions are specific. Congratulations.

REVIEWER COMMENTS

Reviewer #1 (Remarks to the Author):

I thank the authors for their thorough consideration and responses to my first round of remarks. I'll focus these comments around the authors' addressing of my two main points of contention in the first round.

Response: We would like to thank the reviewer for the thorough evaluation of our manuscript and the many useful comments that helped us to improve our work.

1. Concerning the timescale of waning immunity from a single vaccine dose at 8 months, the authors have addressed this via a sensitivity analysis in the supplement and by changing the age endpoint of the main analysis. The sensitivity analysis does demonstrate that the result was highly sensitive to the inclusion of the one seronegative at 28 months of age, and so this result does look more robust by removing them from the sample.

I also thank the authors for adding Figure S4, showing the reader the individual antibody vs. age distributions. At risk of moving the goalposts in a second review, though, this figure reveals a deeper issue - looking at the individual trajectories in Figure S4 reveals that the seronegative child at 28 months who so strongly impacted the waning timescale had apparently never seroconverted to the vaccine in the first place. Removing that child from the analysis of vaccine-derived immunity waning is therefore the right thing to do, as they never had vaccine-derived immunity to wane. This raises the new question for me of whether other non-respondents are present in the sample, and indeed Figure S4 shows that there are. The authors are clear in the text that not every child responds to the vaccine, as expected, but it appears that the children who fail to respond are still being included in the GAM fits that are used to project a timescale for waning of vaccine derived immunity. Failure to seroconvert is a completely separate dynamic from seroconversion followed by waning, and if the goal is to estimate the timescale of waning vaccine immunity, it is absolutely necessary to remove the samples representing children who failed to seroconvert in the first place.

Response: We are grateful to the reviewer for rising this excellent point. We fully agree that the failure to seroconvert after vaccination is a separate dynamic from vaccine-induced seroconversion followed by waning of antibody concentrations. As suggested by the reviewer, we have now removed these observations (n=5) from the main analysis when characterizing the immune response to MCV following different vaccination schedules. Although the estimated dynamics are similar, removing these five observations have remarkably reduced the uncertainty on our estimates. The main text has been updated accordingly:

Lines 176-182. “*Since failure to seroconvert after vaccination is a separate dynamic from vaccine-induced seroconversion followed by waning of antibody concentrations, in the main analysis, we reported MCV-induced antibody response and persistence by excluding individuals who failed to seroconvert after MCV vaccination (n=5) (individual trajectories are shown in **Supplementary Fig. S4**). The Appendix also shows a sensitivity analysis where these observations are not excluded (**Supplementary Fig. S6**).*”

Lines 467-476. “[...], we then used generalized additive mixed models (GAMMs) to fit the log of IgG antibody concentrations at different ages among children who seroconverted after either one dose of MCV1 or the routine two-dose MCV (Appendix, p8). In addition, we performed two sensitivity analyses where we considered the uncertainties in the estimated persistence of MCV-induced antibodies due to individual heterogeneities in the immune response to MCV. In the first sensitivity analysis, we removed random effects in GAMM model; in the second analysis, we included observations from children who failed to seroconvert after MCV vaccination. These sensitivity analyses are reported in Supplementary Figures S6 and S7 in Appendix.”

To show the importance of removing individuals who never seroconverted from the analysis, we have also added a figure in Appendix (Fig. S7, which is also reported below for reviewer’s convenience).

Supplementary Figure S7. Measles-specific antibody dynamics following a two-dose schedule of measles-containing vaccine at 8 and 18 months of age using different statistical models.

Note that the thick curves are the predicted mean value of log-transformed concentrations. Shaded areas show 95% confidence intervals.

2. My second primary point of contention had to do with perceived disagreement between the “proportion susceptible at age A” in Figure 3 and the “cumulative incidence of seroreversion” in Figure 4. The authors’ respond: In this study, when characterizing susceptibility profiles prior to and following different vaccination schedules, we used two different epidemiological outcome measurements: the fraction of susceptible individuals and the cumulative incidence of seroreversion. The fraction of susceptible individuals and the cumulative incidence of seroreversions were obtained with generalized additive models and Kaplan-Meier methods, respectively. In particular, the former quantified the proportion of susceptible individuals at a given age, whereas the latter measured the cumulative occurrence of seroreversions by a given age. As such, for individuals within the same vaccination group, we could observe that their cumulative incidence of seroreversion (Fig. 4) was much higher than the fraction of susceptible individuals (e.g., Fig. 3 and Fig. S5) at a given age. I am unconvinced that the authors’ separation of “proportion susceptible at age A” and “cumulative incidence of seroreversion by age A” resolves this discrepancy. While they are two different quantities, they are not completely independent, they are inextricably linked in a way that this explanation does not resolve – the fraction of susceptible individuals at age A cannot be lower than the cumulative incidence of seroreversion under the conditions here.

More precisely –the fraction of children susceptible at age A is equal to the fraction of children unvaccinated by age A, plus the fraction of children who were vaccinated but failed to seroconvert, plus the fraction of children who seroconverted but seroreverted by age A, minus the fraction of

children who still have their maternal antibodies at age A, minus the fraction of children who got measles before age A. The latter two terms are negligible in this setting, and the first term is zero as the sample only includes vaccinated children. I suppose this assumes that nobody suddenly seroconverts to a vaccine dose received in the distant past, but I think that's a reasonable assumption. So the fraction of children susceptible at age A should be equal to the fraction of non-responders + the fraction of responders who serorevert by age A, and so the cumulative incidence of seroreversion by age A cannot be higher than the fraction of susceptible children at age A.

Having read it a couple of times now, my guess at the root of the issue is that the GAM analysis is failing to address the inherent directional correlation that is induced by using age as an independent variable, and the artifacts that appear because of the censored nature of the data (while the Kaplan-Meier statistic is sort of specifically designed to handle those particular features). Instead, the way the data is aggregated to fit the GAM ignores the fact that the data are repeated measurements on individuals, and that age induces one-way correlations (I think this is less of a problem for the log concentration than for the fraction susceptible). To get concrete – the last 3 bins in Figure 2B contain no susceptibles in the children sampled at those specific ages (18-23 months), so the “Percent susceptible” in each of these these age bins is 0. However, there are 2 susceptibles in the age 16-17 month bin, and 6 in the 14-15 month age bin. Those 8 susceptible kids will still be susceptible when they get to 18-23 months, they just weren't followed up at those time points. The analysis throws out the information value of that natural history picture, and it makes the GAM results unrealistic – Figure 2D (and 3D as well, if you zoom in) predict that the percent susceptible declines as you age further from the vaccination event, which doesn't make sense in the absence of measles transmission.

Response: We fully agree with the reviewer in their assessment of the methodology. To avoid the issue of the GAM model in treating individual's repeated measurements and censored data, in the revised version of the manuscript, we have decided to remove age-specific analyses of susceptibility profiles based on either a descriptive analysis or GAM model. Now we only present the estimated cumulative incidence of seroreversion that were obtained with Kaplan-Meier methods throughout the entire text.

Nonetheless, as none of children with seropositive serum samples after MCV1 vaccination at 8 months of age had shown evidence of seroreversion by the time of their MCV2 vaccination, we were unable to provide long-term susceptibility profiles following a single-dose of MCV at ages 8 months by use of Kaplan-Meier Methods. We have added this limitation to the Discussion:

Lines 329-333. *“Moreover, as none of the children with seropositive serum samples after MCV1 vaccination at 8 months of age had shown evidence of seroreversion by the time of their MCV2 vaccination, we were unable to characterize the long-term susceptibility profiles following a single-dose of MCV at ages 8 months using the Kaplan-Meier method.”*

3. I'm interested in the authors' choice to drop the individual-level random effect when moving from the GLMM model to the GAM model. The justification appears to be in the statement at 154-156: "Furthermore, individual heterogeneities in measles-specific immune response were low, accounting for only 7.8% of the total variance". I would argue that the entire model only explained 40% of the variance, so the individual heterogeneity accounts for 20% of the model's explanatory power, hardly negligible.

Response: We would like to thank the reviewer for this comment. Following reviewer's suggestion, in the current version of the manuscript, we assessed the goodness-of-fit for generalized additive model (GAM) and generalized additive mixed model (GAMM) using the same observations from individuals who seroconverted after MCV vaccination. These two models included the same covariates for measles-specific antibody concentrations (i.e., age and vaccination status), while the latter (GAMM model) additionally included random effects that allowed individual heterogeneities in immune response to MCV. The comparison of several goodness-of-fit statistics for GAM/GAMM models and corresponding dynamic of measles-specific antibody concentration are presented in Table S4 and Fig. S7, respectively. We found that the inclusion of random effects could significantly improve goodness-of-fit measure in the kinetic model of antibody decay (Table. S4). Therefore, in the current version of manuscript, we re-estimated the dynamics of measles-specific antibody concentration prior to and following different vaccination schedules by use of GAMM models instead of the original GAM models. The inclusion of random effects in the main analysis has been described in the following paragraph:

Lines 153-157: "Furthermore, although individual heterogeneities in measles-specific immune response were relatively low (7.8% of the total variance), random effects that allowed to account for individual heterogeneities in immune response to MCV were introduced in the kinetic model of antibody decay (Supplementary Table S4)."

4. I would appreciate if the authors would please clarify the ethical concerns that prevent making the disaggregated data publicly available. The aggregated data made available in Github (as a side note, the link to the Github repo) appear to only allow someone to plot Figs 2A, 3A, and 4; it does not allow for any meaningful check on the analysis presented herein or to subject the data to further scrutiny. These data are being used in the manuscript to argue for changes to vaccination schedules, a topic with global implications for health, and it seems to me that the components of the raw data critical to that discussion can be provided in a way that protects patient privacy - e.g., removing patient identifying information and demographic characteristics but still providing the disaggregated IgG titer measurements and months at which they were taken, with individual linkage made possible by an anonymous ID. Minimally, what a reader would need to replicate the Supplementary Figure 4; ideally, also including the trajectories for children not shown in Figure S4 because they were vaccinated at

non-standard ages.

Response: As suggested, we have now provided individual observations (with anonymous ID) prior to and following different vaccination schedules in Github. This data allows replicating all the analyses.

Reviewer #2 (Remarks to the Author):

I would like to congratulate the authors on the revised manuscript. I believe that the changes made more than adequately address all of the comments provided. There were a few minor grammatical edits that I noted:

Response: We would like to thank the reviewer for their positive assessment of our manuscript.

1. Abstract line 43: I believe the word "that" at the end of the line is not needed.

Response: Thank you, correction made.

2. Introduction line 73: Would it be more clear if you used the phrase "due to" or "caused by" rather than "given an"?

Response: Thank you, correction made.

3. line 81: there's a long phrase but I think it should be the "recommended age...has" instead of "have".

Response: The reviewer is correct. Thank you for the careful reading.

4. line 91: Should "and interfere" be changed to "without interfering"?

Response: Thank you again for the careful reading.

5. lines 114, 116, 247, and 364: I would suggest to use the word "doses" rather than "cycle".

Response: We revised the text throughout the manuscript as suggested by the reviewer.

6. line 208 consider "following MCV2" rather than "of MCV2"

Response: Thank you, correction made.

7. lines 248-249: I would suggest to re-phrase "concentrations and increases of" to "concentrations which leads to increase in susceptibility", which I believe is more clear.

Response: In light of Reviewer #1's comments, this sentence has been removed.

8. line 272: Consider changing "things that need to be considered" to "considerations"

Response: Thanks, corrections made.

Otherwise, the manuscript looks great. The analysis looks great and the conclusions are specific.

Congratulations.

Response: We would like to thank the reviewer once again for their positive assessment of our work.

REVIEWERS' COMMENTS

Reviewer #1 (Remarks to the Author):

I would like to thank the authors for a thoughtful dialogue and thorough responses to each of my comments. All of my concerns have been addressed - the data analysis and conclusions are clear and relevant to vaccination policy. Congratulations.